# The CXCL8/MAPK/hnRNP-K axis enables susceptibility to infection by EV-D68, rhinovirus, and influenza virus in vitro

Qingran Yang [1,2], Haoran Guo[2], Huili Li[2], Zhaoxue Li[2], Fushun Ni[2], Zhongmei Wen[3], Kai Liu [4], Huihui Kong[5,6] & Wei Wei [2,7] ✉

Respiratory viruses pose an ongoing threat to human health with excessive cytokine secretion contributing to severe illness and mortality. However, the relationship between cytokine secretion and viral infection remains poorly understood. Here we elucidate the role of *CXCL8* as an early response gene to EV-D68 infection. Silencing CXCL8 or its receptors, CXCR1/2, impedes EV-D68 replication in vitro. Upon recognition of CXCL8 by CXCR1/2, the MAPK pathway is activated, facilitating the translocation of nuclear hnRNP-K to the cytoplasm. This translocation increases the recognition of viral RNA by hnRNP-K in the cytoplasm, promoting the function of the 5′ untranslated region in the viral genome. Moreover, our investigations also reveal the importance of the CXCL8 signaling pathway in the replication of both influenza virus and rhinovirus. In summary, our findings hint that these viruses exploit the CXCL8/MAPK/hnRNP-K axis to enhance viral replication in respiratory cells in vitro.

During viral infection, the immune system is tasked with a delicate balance. It must mount a robust response to control viral replication while avoiding the overproduction of cytokines, which can lead to a cytokine storm and exacerbate tissue damage[1,2]. Cytokines, such as interferons and interleukins, are pivotal in stimulating the production of antiviral proteins and modulating immune responses[3]. However, viruses have evolved strategies to counteract these defenses by modulating cytokine signaling pathways[4,5]. This parasitic relationship with the host necessitates a deep understanding of how viruses exploit host resources and manipulate cellular processes for their propagation.

Infection with respiratory enterovirus D68 (EV-D68), a member of the Picornaviridae family[6], results primarily in respiratory symptoms and can potentially lead to respiratory failure[7–10]. EV-D68 is a matter of concern due to recent outbreaks and accumulating evidence suggesting that infection with EV-D68 can result in a severe polio-like

neurological complication known as acute flaccid myelitis[7,9]. However, no drugs targeting enteroviruses are currently available. Improving our understanding of viral manipulation of host cells and the transmission of proviral signals is of significant importance.

Chemokines play crucial roles in various biological processes including angiogenesis, tumor growth and metastasis, hematopoiesis, organogenesis, cell survival, differentiation, and are intimately associated with the pathogenesis of infectious diseases[11]. C-X-C motif chemokine ligand 8 (CXCL8), also known as Interleukin-8 (IL-8), is a chemokine that plays a critical role in the recruitment and activation of neutrophils during inflammatory responses. CXCL8 primarily interacts with its receptors, CXCR1 and CXCR2, which are predominantly expressed on neutrophils but are also found on other myeloid and lymphoid immune cells. Additionally, these receptors are present on non-leukocytes such as fibroblasts, neurons, astrocytes, endothelial

[1]Department of Respiration, Children's Medical Center, First Hospital, Jilin University, Changchun, Jilin, China. [2]Institute of Virology and AIDS Research, First Hospital, Jilin University, Changchun, Jilin, China. [3]Center for Pathogen Biology and Infectious Diseases, Department of Respiratory Medicine, First Hospital, Jilin University, Changchun, Jilin, China. [4]Department of Chemistry, Tsinghua University, Beijing, China. [5]Influenza Research Institute, Department of Pathobiological Sciences, School of Veterinary Medicine, University of Wisconsin-Madison, Madison, WI, USA. [6]State Key Laboratory of Veterinary Biotechnology, Harbin Veterinary Research Institute, the Chinese Academy of Agricultural Sciences, Harbin, China. [7]Cancer Center, Key Laboratory of Organ Regeneration and Transplantation of Ministry of Education, Institute of Translational Medicine, First Hospital, Jilin University, Changchun, Jilin, China. ✉e-mail: wwei6@jlu.edu.cn

cells, epithelial cells, and hepatocytes[12]. The activation of these receptors promotes the migration of immune cells to sites of infection or inflammation, making the CXCL8/CXCR1-2 axis crucial in the host response to infections. Moreover, CXCL8 has been demonstrated to regulate the migration of non-immune cells and stimulate corneal neovascularization in vivo[13].

The expression of the CXCL8 gene is locally upregulated following receptor activation by pro-inflammatory stimuli, such as IL-1 or TNF-α, in response to inflammatory triggers[14,15]. Elevated levels of CXCL8 have been documented in various inflammatory diseases, including rheumatoid arthritis and inflammatory bowel disease, thereby contributing to sustained inflammation and tissue damage[16]. CXCL8 is upregulated in various cancers, where it can promote tumor growth, invasion, angiogenesis, metastasis, and drug resistance through autocrine or paracrine mechanisms. Recent studies have also shown that CXCL8 can modulate the immune microenvironment, influencing the efficacy of immune checkpoint inhibitors in cancer treatment[17,18]. However, studies exploring the relationship between CXCL8 and viral replication capacity remain notably limited.

Heterogeneous nuclear ribonucleoproteins (hnRNPs) represent a class of nuclear proteins that are integral to a multitude of cellular processes, including transcription, post-transcriptional modification, and the maturation of precursor mRNA (pre-mRNA). Beyond these functions, hnRNPs are pivotal in modulating mRNA stability and orchestrating its nucleocytoplasmic transport[19]. Among the hnRNP family, hnRNP-K is distinguished by its pronounced and stable binding affinity for poly(C) sequences. hnRNP-K is characterized by the absence of canonical RNA-binding motifs but is endowed with three KH domain repeats[20,21]. hnRNP-K plays a critical role in the replication of multiple viruses, including hepatitis B virus (HBV), cytomegalovirus (CMV), herpes simplex virus-1 (HSV-1), hepatitis C virus (HCV), and influenza A virus (IAV)[22–27]. However, the function of hnRNP-K as either a pro-viral or anti-viral factor varies significantly across different viral systems. Previous studies have demonstrated that hnRNP-K interacts with the 5' untranslated region (5'UTR) of enterovirus A71 (EV-A71) and is essential for viral replication[28]. Nevertheless, the precise molecular mechanisms by which enteroviruses, which complete their entire replication cycle in the cytoplasm, exploit the nuclear-localized hnRNP-K protein to facilitate their own replication remain poorly understood.

In this work, we explore the role of CXCL8 in the context of EV-D68 infection. Unlike other cytokines that play a role in antiviral signaling, CXCL8 facilitates EV-D68 infection, which supports the function of the viral 5'UTR by promoting the cytoplasmic accumulation of hnRNP-K and its binding to viral RNA (vRNA). This mechanism is not unique to EV-D68, as several human respiratory viruses, including influenza virus, rhinoviruses, and SARS-CoV-2, upregulate the expression of CXCL8 in respiratory cells to enhance viral replication. Our study sheds new light on the complex interplay between viruses and inflammatory cytokines and underscores the potential of targeting CXCL8-related signaling cascades as a therapeutic strategy for respiratory virus infections.

## Results

### CXCL8 production is an early host response to EV-D68 infection
To identify early host responses that modulate viral infection, we analysed the effects of low-dose viral infection (EV-D68, multiplicity of infection (MOI) = 0.1) on the whole transcriptome of A549 respiratory cells. Samples were collected for RNA-Seq analysis at an early stage of infection (4 h post-infection (hpi)) to mitigate the extensive variations in host gene expression induced by multiple rounds of viral replication. The CXCL8 gene, which encodes the C-X-C motif chemokine ligand 8, was the most significantly upregulated gene in all three replicate EV-D68-infected samples compared with the uninfected samples (Fig. 1a). This finding was further validated by direct detection of CXCL8 mRNA via real-time PCR, which confirmed the continuous

increase in the CXCL8 transcript abundance during EV-D68 infection, with no significant difference observed in the control groups (Fig. 1b). CXCL8, also called Interleukin-8/IL8, is a secreted chemokine that plays a crucial role as a major mediator of the inflammatory response. Therefore, we further assessed the amount of secreted CXCL8 protein in the cell supernatant following infection with EV-D68 at MOIs of 0.03 or 0.12 (Fig. 1c). As expected, the data revealed greater accumulation of secreted CXCL8 during EV-D68 infection than in the noninfected groups (Fig. 1c). These results were further supported by data showing that EV-D68 infection increased the transcription of CXCL8 in primary isolated human bronchial epithelial cells (HBECs) (Fig. 1d). Hence, CXCL8 is an early response gene for EV-D68 infection in human respiratory epithelial cells.

### CXCL8 is required for EV-D68 replication
Our finding that the early post-viral infection response specifically triggers the production and secretion of CXCL8 prompted us to further investigate the role of CXCL8 in EV-D68 replication. After lentiviral transduction of short hairpin RNA (shRNA) targeting CXCL8 mRNA (Fig. 2a), the increase in the abundance of CXCL8 proteins caused by EV-D68 infection was inhibited (Fig. 2b). Interestingly, stable knockdown of CXCL8 alleviated cytopathic effects (CPEs) in EV-D68-infected cells compared with control cells (Fig. 2c). CXCL8 downregulation also significantly impaired the accumulation of viral RNA in both infected cells (Fig. 2d) and culture supernatants (Fig. 2e) compared with that in the control groups. Furthermore, the synthesis of the viral structural protein VP1 (Fig. 2f–h) and the titer of EV-D68 progeny virions (Fig. 2i) were substantially lower in CXCL8-knockdown cells than in control cells. We further confirmed the essential role of CXCL8 expression in EV-D68 infection in diverse immortalized cell lines, including A549, BEAS-2B, and CALU-3 respiratory cells and other EV-D68-permissive cell lines, namely, HEK293T cells and RD cells (Supplementary Figs. 1 and 2). Decreasing CXCL8 expression greatly decreased virus-associated CPEs and the progeny virus titers of EV-D68 and circulating isolated strains (MO and KY) (Supplementary Figs. 1 and 2). In addition to immortalized cells, CXCL8 was found to be required for EV-D68 replication also in primary HBECs (Fig. 2j, k, Supplementary Fig. 3). We further elucidated the role of CXCL8 in EV-D68 replication within airway organoids by employing a CXCL8 neutralizing antibody, MAB208 (R&D Systems), to functionally inhibit CXCL8. Our results demonstrated that treatment with the anti-CXCL8 antibody significantly decreased the replicative capacity of EV-D68 in airway organoids (Fig. 2l). Hence, CXCL8 expression is required for EV-D68 infection.

We next performed virus attachment and entry assays and confirmed that the expression of CXCL8 does not influence EV-D68 entry into host cells (Supplementary Fig. 4a, b). The one-step growth curve generated to assess viral RNA replication further revealed that CXCL8 downregulation inhibited the accumulation of EV-D68 RNA at the early time point of 4 hpi (Supplementary Fig. 4c), suggesting that CXCL8 contributes to the early post-entry stages of EV-D68 infection. We then used a well-established reporter system to evaluate the activity of the enteroviral 5'UTR. The EV-D68 minireplicon contained the luciferase reporter gene flanked by UTRs derived from EV-D68 RNA and was inserted into vectors with the RNA polymerase I promoter[29]. Following transfection, luciferase signals were detectable in the cells. We ultimately confirmed that CXCL8 expression is crucial for the function of the 5'UTR (Fig. 2m). Silencing of CXCL8 inhibited the activity of only the EV-D68 5'UTR; no effect on the HIV-1 5'LTR was observed (Supplementary Fig. 5).

### CXCL8-CXCR1/2 signaling facilitates EV-D68 replication
CXCL8, a well-known cytokine, facilitates the transmission of corresponding cellular signals by being secreted into the extracellular space. By employing exogenous recombinant human CXCL8 (rCXCL8) protein, we found that supplementation with increasing

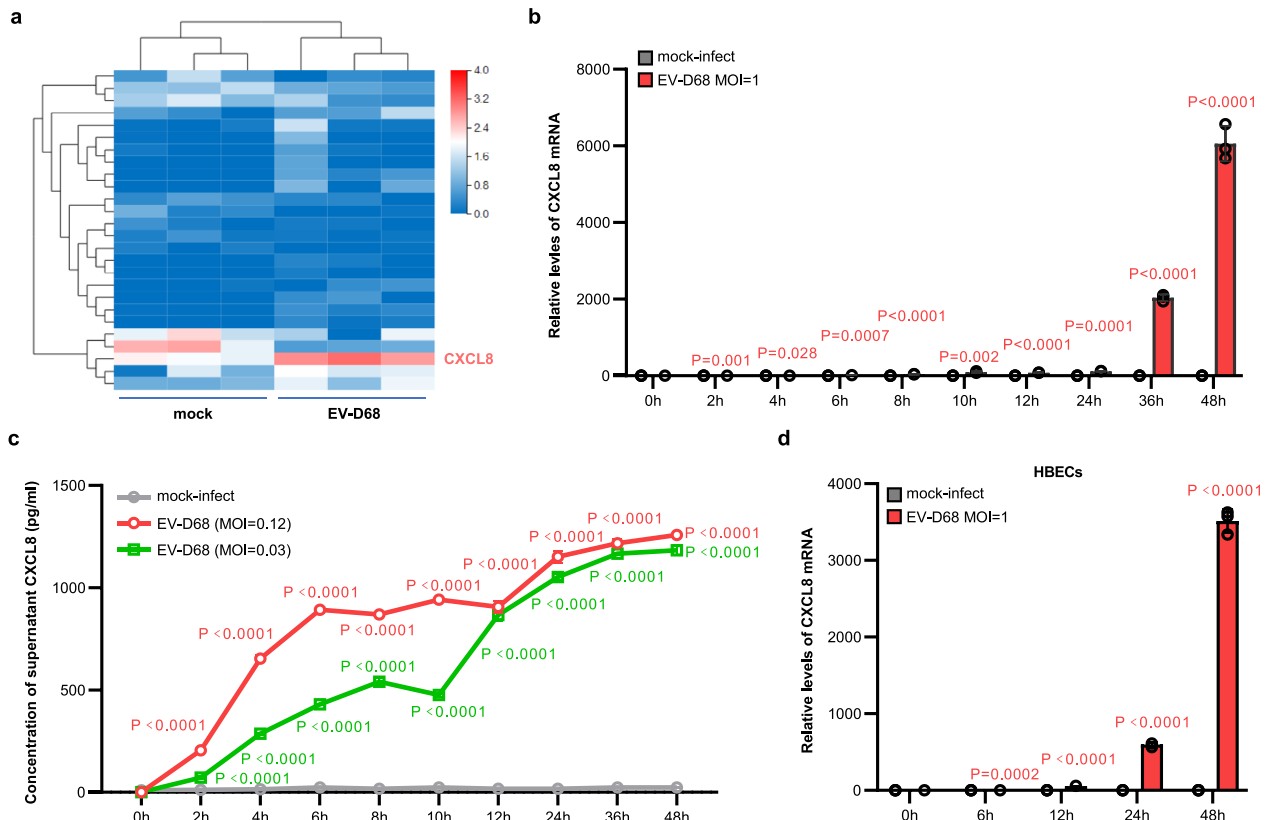

**Fig. 1 | EV-D68 infection activates CXCL8 expression in human respiratory cells.**
**a** Heatmap visualization of the clustering of differentially expressed genes between EV-D68 (MOI = 0.1)-infected A549 cells and mock-infected cells at 4 h post-infection (hpi). Three replicate samples were set up in each group. **b, c** Significant increases in CXCL8 transcription and secretion were detected upon EV-D68 infection. A549 cells infected or mock-infected with EV-D68 were collected at the indicated time points. The relative *CXCL8* mRNA level was measured via qRT–PCR (**b**), and the concentration of secreted CXCL8 was determined via ELISA (**c**). **d** Marked upregulation of *CXCL8* transcription in human bronchial epithelial cells (HBECs) after EV-D68 infection (MOI = 1). Infected or mock-infected cells at 0 h in (**b–d**) were set as the control groups. $N$ = 3 (**a–d**) biological replicates. Data are represented as mean ± SD. Two-tailed *t*-test (**b** and **d**) or two-way ANOVA (**c**) was used to assess statistical significance. Source data are provided as a Source Data file.

concentrations of the CXCL8 protein effectively restored the replication ability of EV-D68 in CXCL8-knockdown cells (Fig. 2n and Supplementary Fig. 6).

CXCL8 recognizes and activates the G protein-coupled receptors (GPCRs) CXCR1/2, thereby inducing the activation and trafficking of inflammatory mediators and facilitating tumour progression, invasion, and metastasis. Thus, we investigated the role of CXCR1/2 in regulating EV-D68 replication mediated by CXCL8. To this end, we generated A549 cell lines with stable knockdown of either CXCR1 or CXCR2 and exposed them to EV-D68 while simultaneously infecting A549-shctl and A549-shCXCL8 cell lines. Compared with the corresponding control cells, both the shCXCR1 and shCXCR2 cells, similar to the shCXCL8 cells, presented significant attenuation of CPEs (Fig. 3a), VP1 synthesis (Fig. 3b, c), and progeny virion production (Fig. 3d) associated with EV-D68 infection.

The CXCR1/2 receptors have approximately 76% sequence homology and play a pivotal role in the initiation and spread of inflammatory processes and are associated with tumor growth and metastasis by binding to their ligand CXCL8[30]. Several small molecule inhibitors targeting CXCR1/2 have undergone clinical trials for their antitumor properties, thereby establishing a valuable repertoire of candidate drugs that exhibit safety and efficacy in human subjects[31–33]. Exploiting the advantages revealed in these trials, we discovered that the CXCR1/CXCR2 antagonist SX682 (currently in phase I/II clinical trials for metastatic colorectal adenocarcinoma) and the CXCR2-specific inhibitor Danirixin (currently entering phase II clinical trials for pulmonary disease) effectively inhibited EV-D68 RNA replication (Fig. 3e and j). Further studies demonstrated dose-dependent

inhibition of viral RNA replication by SX682 and Danirixin (Fig. 3f and k). On the basis of the proliferation curve generated to assess RNA replication, we calculated the half-maximal effective concentrations ($EC_{50}$) against EV-D68 at 36 hpi for SX682 (7.522 μM) and Danirixin (7.82 μM). Cell viability assays revealed no impact on cell viability at concentrations lower than 30 μM for either drug (Supplementary Fig. 7c, d), ruling out the possibility of any cytotoxic interference from these compounds. Treatment with either SX682 or Danirixin effectively suppressed CPEs (Supplementary Fig. 7a, b), VP1 synthesis (Fig. 3g, h and l, m), and viral 5'UTR activity (Fig. 3i and n), consistent with the aforementioned findings. In addition, SX682 exhibited potent anti-EV-D68 activity in a variety of respiratory cells and other EV-D68-permissive cells (HEK293T and RD cells) (Supplementary Fig. 8). These findings highlight the targeting of the CXCL8 receptors CXCR1/CXCR2 as an ideal approach for inducing an antiviral response against EV-D68.

## CXCL8-induced proviral effects rely on the activation of MEK-ERK signaling

Upon the binding of CXCL8 to its receptor CXCR1/2 and the subsequent activation of intracellular effectors, diverse cellular responses, including chemotaxis, cell adhesion, angiogenesis, the release of proinflammatory cytokines, and the modulation of immune responses, are elicited[30]. We subsequently investigated the involvement of downstream effectors of CXCL8 in EV-D68 infection and identified significant associations between these processes and the mitogen-activated protein kinase (MAPK) pathway. First, EV-D68 infection

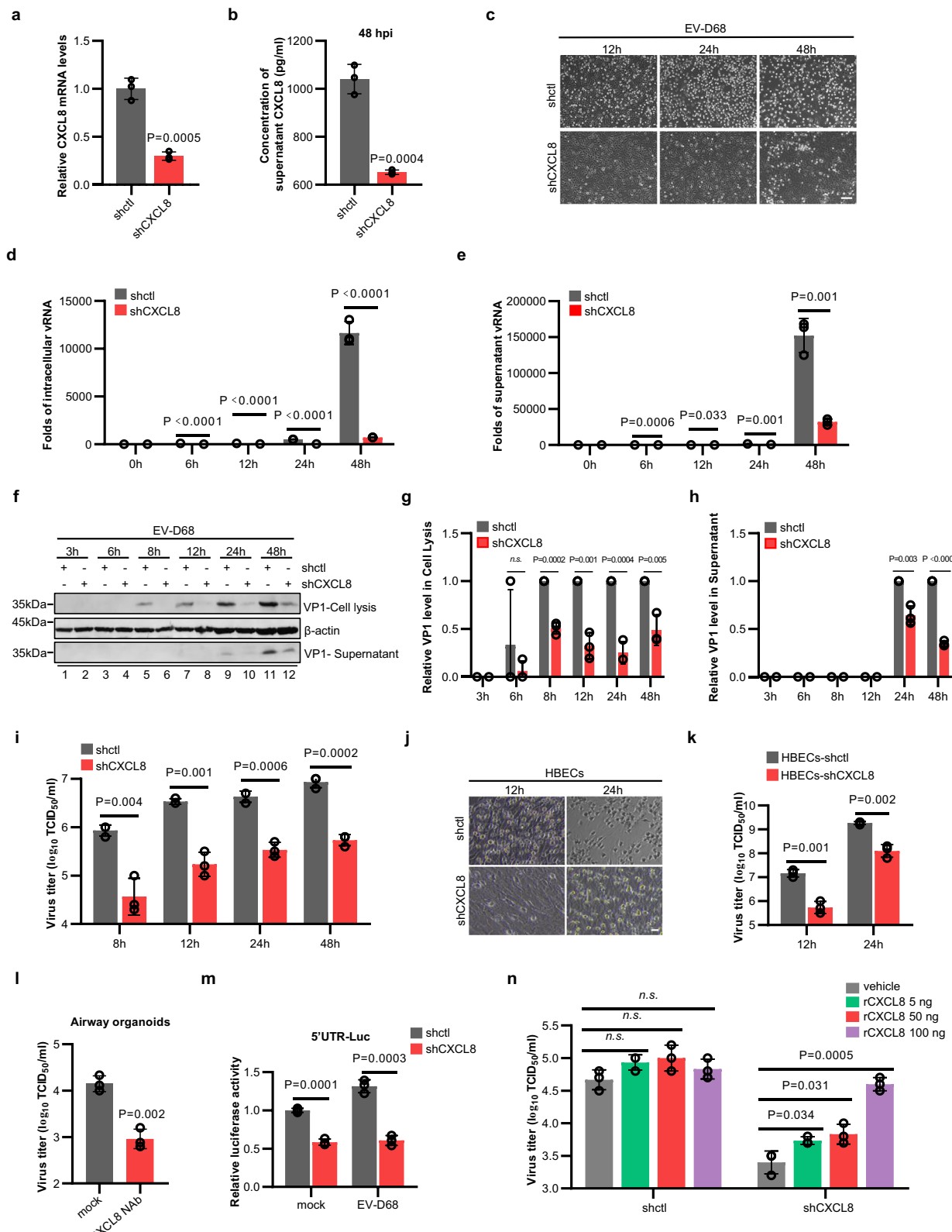

significantly increased the phosphorylation of MEK1/2 and ERK1/2, which are hallmarks of MAPK signaling activation (Fig. 4a–c). Moreover, treatment with rCXCL8 resulted in increased activation of MAPK signaling even in the absence of viral infection (Supplementary Fig. 9a, b). Downregulating the expression of CXCL8 or its receptors impaired EV-D68-facilitated MAPK activation (Fig. 4a–c and Supplementary Fig. 9c, d). Consistent with these findings, knockdown of either MEK1 or

MEK2 blocked EV-D68-induced ERK1/2 phosphorylation (Fig. 4d–g), confirming that EV-D68 triggers CXCL8-CXCR1/2-MEK-ERK signal transduction. To elucidate the role of CXCL8 in activating the MAPK signaling pathway to facilitate EV-D68 replication, we employed an agonist of the MAPK pathway, namely, C16-PAF (Supplementary Fig. 9e, f). Notably, treatment with C16-PAF effectively restored EV-D68 replication in CXCL8-silenced cells (Supplementary Fig. 9g, h).

**Fig. 2 | CXCL8 promotes EV-D68 replication. a** Knockdown efficiency of CXCL8 in A549-shCXCL8 cells, as determined by qRT–PCR. **b** EV-D68 (MOI = 0.01) was used to infect A549-shctl and A549-shCXCL8 cells, and the concentration of CXCL8 in the supernatant was measured at 48 h post-infection by ELISA. **c–i** Silencing of CXCL8 inhibits EV-D68 replication. A549-shctl and A549-shCXCL8 cells were infected with EV-D68 (MOI = 0.01). Cytopathic effects (**c**), the viral RNA abundance in cells (**d**) and in the supernatant (**e**), the expression of the viral structural protein VP1 (**f**) and the viral titer (**i**) were measured at different time points post-infection. **g, h** Quantitative analysis of the relative VP1 protein levels in (**f**). **j, k** The replication of EV-D68 in HBECs requires the presence of CXCL8. HBEC-shctl and HBEC-shCXCL8 cells were infected with EV-D68 at an MOI of 0.01. CPEs (**j**) and the viral titer (**k**) were measured after 12 or 24 h. **l** Inhibition of CXCL8 activity attenuates EV-D68 replication in airway organoids. Human airway organoids differentiated for 28 days were infected with EV-D68 ($5.0 \times 10^5$ TCID$_{50}$) in the presence or absence of 20 ng/ml CXCL8 neutralizing antibody (MAB208). The viral titer in the medium was measured 48 h post-infection. **m** CXCL8 is critical for EV-D68 5′UTR activity. A549-shctl and A549-shCXCL8 cells were co-transfected with pol I-EV68 5′UTR-Luc and Renila luciferase, and separate groups were established for EV-D68 infection and mock infection. Luciferase activity was measured 24 h post-transfection. **n** Recombinant CXCL8 (rCXCL8) restores the replication capacity of EV-D68 in A549-shCXCL8 cells. $N = 3$ (**a, b, d, e, g–i, k–n**) biological replicates. Representative of three biologically independent experiments (**c, f, j**). Data are represented as mean ± SD. Two-tailed $t$-test (**a, b, d, e, g–i, k–n**) was used to assess statistical significance. Scale bar in (**c** and **j**) is 100 μm. Source data are provided as a Source Data file.

The replication capacity of EV-D68 in A549 cells with stable knockdown of MEK1 or MEK2 was subsequently investigated and compared with that in control cells. The results revealed significant attenuation of viral replication in a time-dependent manner upon silencing of either MEK1 or MEK2 (Fig. 4h, i). Targeting of the MAPK pathway remains a promising area in the development of antitumor drugs, and numerous small molecule inhibitors exhibiting significant selectivity for MEK1/2 have been identified[34]. Indeed, we determined that treatment with noncytotoxic concentrations of the MEK1/2 inhibitor U0126 results in significant resistance to EV-D68 infection in respiratory cells (Fig. 4j–n). The impact of U0126 on the viral replication cycle primarily involves interference with viral 5′UTR activity, leading to the blockade of viral RNA replication and protein translation (Fig. 4o). In further validation of these findings, we also discovered that SX682 and U0126 effectively inhibited EV-D68 replication in human airway organoids (Fig. 4p).

### CXCL8 signaling enhances the recognition of viral RNA by hnRNP-K

Our findings progressively revealed that EV-D68 specifically stimulates CXCL8 expression and activates associated pathways to maintain the viral 5′UTR in a relatively active state. The enteroviral 5′UTR functions as an internal ribosome entry site (IRES) element, contributing to the modulation of viral pathogenesis through its interaction with host factors and its impact on viral RNA replication and translation efficiency[35,36]. To elucidate the factors interacting with the viral 5′UTR that are involved in the CXCL8-mediated promotion of viral replication, we validated known viral RNA-interacting factors (PCBP1, PCBP2, and hnRNP-K). While silencing CXCL8 did not affect the interaction between either PCBP1 or PCBP2 and viral RNA (Supplementary Fig. 10a–d), the intracellular localization of the host factor hnRNP-K, which plays a critical role in multiple aspects of viral RNA replication and protein translation, was significantly influenced by the CXCL8 signaling pathway (Supplementary Fig. 10e).

The results of the nuclear–cytoplasmic fractionation assay revealed that knockdown of CXCL8 significantly attenuated the accumulation of the hnRNP-K protein in the cytoplasmic fraction of EV-D68-infected cells, leading to predominant nuclear localization of the hnRNP-K protein (Fig. 5a, b). Moreover, immunofluorescence (IF) analysis was used to evaluate hnRNP-K protein localization at 0, 4, 6, and 8 h post virus infection. Notably, under uninfected conditions, hnRNP-K is localized predominantly in the nucleus; however, upon EV-D68 infection, gradual translocation of the hnRNP-K protein to the cytoplasm was observed during the early stages of infection (Fig. 5c). Importantly, knocking down CXCL8 expression resulted in sustained nuclear retention of hnRNP-K even at 6 hpi (Fig. 5c). Since enterovirus replication occurs exclusively in the cytoplasmic compartment, nucleus-localized hnRNP-K lacks spatial opportunities for binding viral RNA. Our coimmunoprecipitation data further demonstrated a significant reduction in the interaction between viral RNA and hnRNP-K following CXCL8 knockdown (Fig. 5d–f). Additionally,

immunofluorescence analysis revealed that silencing the downstream signaling factor MEK1 or MEK2 within the CXCL8 pathway impeded the viral infection-induced translocation of the hnRNP-K protein to the cytoplasm (Fig. 5g). Consistent with these results, treatment with the MAPK inhibitor U0126 also effectively suppressed the rCXCL8-induced cytoplasmic translocation of hnRNP-K in the absence of viral infection (Supplementary Fig. 10f). In summary, our findings underscore the role of CXCL8 signaling pathway activation in promoting the cytoplasmic accumulation of hnRNP-K proteins and their binding to viral RNA.

### The viral structural protein VP4 activates CXCL8 signaling

After understanding the role of CXCL8 in facilitating viral replication in respiratory cells, we shifted our focus to the upstream pathway, seeking to elucidate the mechanism by which the virus triggers CXCL8 expression. Initially, we conducted unbiased screening to assess the impact of viral proteins on the intracellular *CXCL8* mRNA level. Notably, the expression of the viral structural protein VP0 alone was sufficient to activate CXCL8 expression (Fig. 6a). Furthermore, through the progressive increase in its protein expression, VP0 clearly upregulated CXCL8 expression in a dose-dependent manner (Fig. 6b). As an integral component of the virion, VP0 undergoes cleavage during virion maturation, resulting in the production of two distinct structural proteins: VP2 and VP4[37]. Our subsequent results revealed that VP4, the cleavage product of VP0, plays a pivotal role in increasing the intracellular *CXCL8* mRNA level (Fig. 6c). By establishing a luciferase reporter system based on the promoter sequence of CXCL8, we successfully demonstrated that both VP0 and VP4 can directly drive *CXCL8* promoter activity (Fig. 6d). These findings provide compelling evidence clarifying that the utilization of virion components by EV-D68 induces CXCL8 expression and activates associated signaling pathways during the early stages of viral invasion.

Myristoylation of enterovirus VP0/VP4 in host cells is essential for maintaining its functional role in viral infection[38,39]. In this study, we compared the myristoylation-deficient VP0 mutant (VP0-G2A) with wild-type VP0 and confirmed the crucial role of myristoylation in the VP0-induced upregulation of CXCL8 expression (Supplementary Fig. 11), thereby further supporting the specific function of VP0/VP4 molecules in inducing CXCL8 expression.

### VP4-induced CXCL8 expression is dependent on the syk-PI3K-AKT pathway

Bioinformatics analysis of the VP4 sequence and spatial structure revealed the presence of an evolutionarily conserved "YXXL/I" motif in the VP4 protein, which serves as a binding motif for the activation of the syk-PI3K signaling pathway in cellular processes[40]. Consequently, we validated the importance of this motif in the functions of VP4 by demonstrating that mutations (Y27A or I30S) within this region, in either VP0 or VP4, can disrupt viral protein functions, leading to CXCL8 activation (Fig. 6e–h).

Considering the crucial role of the syk-binding motif in CXCL8 induction, we demonstrated that EV-D68 infection could increase AKT

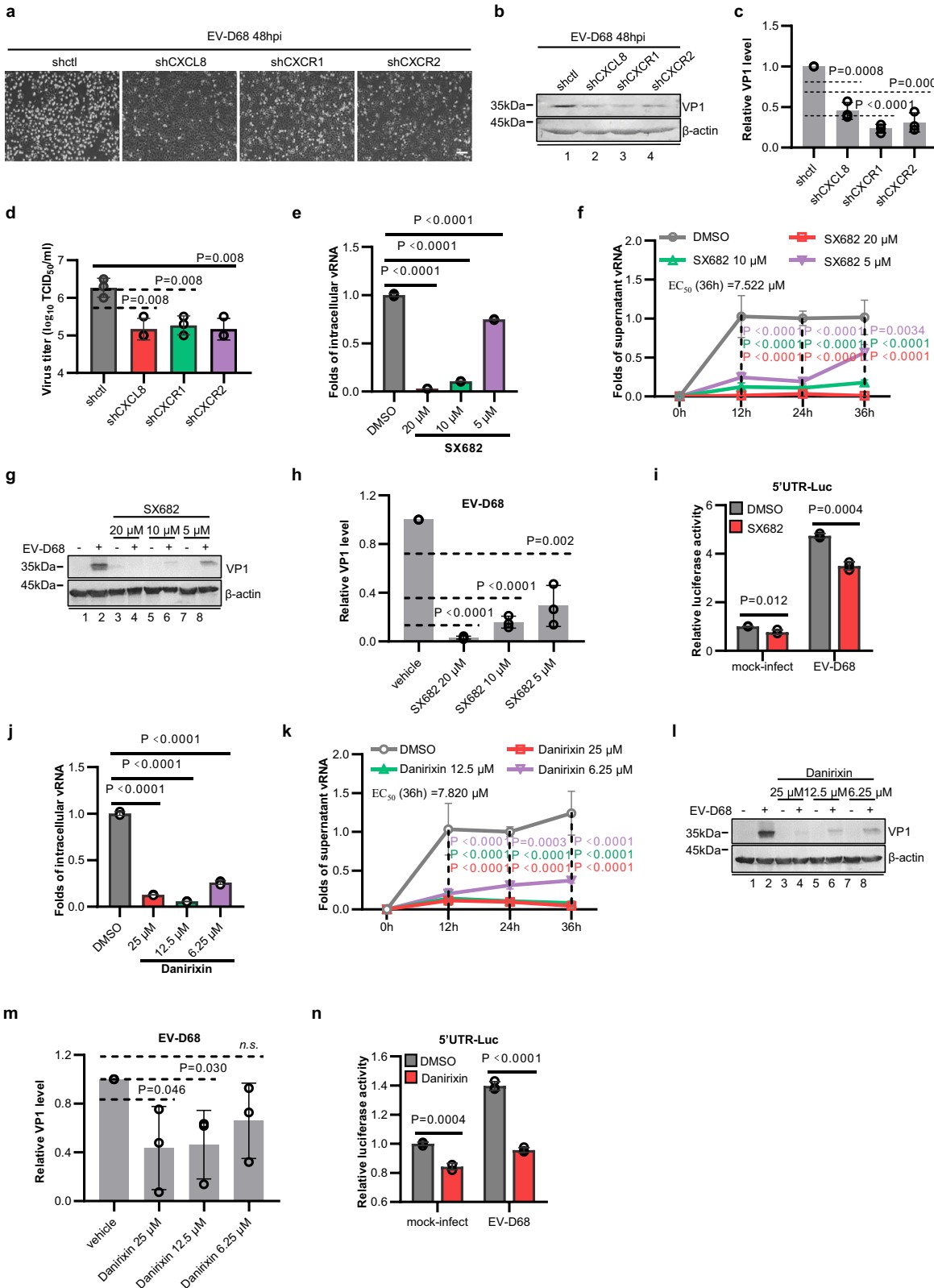

phosphorylation at Ser473 (Supplementary Fig. 12a, b). Moreover, the knockdown of endogenous syk impaired this increase (Fig. 6i, j). These findings prompted us to investigate the involvement of the syk-PI3K-AKT axis in EV-D68 VP4-mediated CXCL8 induction. To this end, cells with stable knockdown of syk, PIK3CA, PIK3CB, or AKT were transfected with VP4 expression vectors or challenged with an equal dose of EV-D68. The *CXCL8* mRNA level was measured 24 h later. Our repeated

experiments consistently indicated that the presence of syk, PIK3CA, and AKT−but not PIK3CB−is necessary for the VP4/EV-D68-induced increase in CXCL8 expression (Fig. 6k and Supplementary Fig. 12c). Silencing these genes also disrupted downstream ERK activation (Supplementary Fig. 12d, e) and significantly reduced the viral replication capacity (Fig. 6l), providing further support for our conclusion that the viral structural protein VP4 participates in a syk-PI3K-AKT

**Fig. 3 | CXCL8-CXCR1/2 signaling facilitates EV-D68 replication.**
**a**, **b**, **d** Knockdown of CXCR1/2 suppresses EV-D68 replication in A549 cells. A549-shctl, A549-shCXCL8, A549-shCXCR1, and A549-shCXCR2 cells were infected with EV-D68 (MOI = 0.01), and CPEs (**a**), viral protein VP1 expression (**b**), and the viral titer (**d**) were measured 48 h after infection. **c** Quantitative analysis of the relative VP1 protein levels in (**b**). **e**–**g** Treatment with the CXCR1/2 inhibitor SX682 inhibits EV-D68 replication. **h** Quantitative analysis of the relative VP1 protein levels in (**g**). **i** SX682 inhibits the activity of the EV-D68 5′UTR. **j**–**l** Treatment with the CXCR2

antagonist Danirixin inhibits EV-D68 replication. **m** Quantitative analysis of the relative VP1 protein levels in (**l**). **n** Danirixin inhibits the activity of the EV-D68 5′UTR. Cells treated with DMSO at 12 h, 24 h or 36 h post-infection in (**f** and **k**) were set as the control groups. $N = 3$ (**c**–**f**, **h**–**k**, **m**, **n**) biological replicates. Representative of three biologically independent experiments (**a**, **b**, **g**, **l**). Data are represented as mean ± SD. Two-tailed $t$-test (**c**–**e**, **h**–**j**, **m**, **n**) or two-way ANOVA (**f** and **k**) was used to assess statistical significance. Scale bar in (**a**) is 100 μm. Source data are provided as a Source Data file.

signaling cascade to increase *CXCL8* gene expression and cytokine release for the transmission of proviral signals to not only infected cells but also bystander cells.

## The requirement for CXCL8 signaling is shared by human respiratory viruses

Interestingly, we observed significant homology between the VP4 protein of the respiratory virus human rhinovirus (HRV), which belongs to the enterovirus family, and the VP4 protein of EV-D68. Both proteins contained a conserved "YXX(X)I" sequence. Moreover, HRV infection increased the expression of CXCL8 (Fig. 7a). Consequently, we investigated the functionality of the HRV VP4 protein and discovered its similarity to that of EV-D68 VP4. Notably, the HRV VP4 protein also increased *CXCL8* promoter activity (Fig. 7b, c). Furthermore, a mutation in the syk-interacting motif (Y27A) impaired the capacity of HRV VP4 to upregulate CXCL8 expression (Fig. 7d, e). In addition, HRV infection activated the MAPK pathway in a CXCL8-dependent manner (Supplementary Fig. 13a, b). Correspondingly, the inhibition of CXCL8 also resulted in a significant reduction in the HRV replication efficiency (Fig. 7f). Mechanistically, HRV induced the activation of the CXCL8 signaling pathway to facilitate the translocation of the nuclear protein hnRNP-K into the cytoplasm, thereby increasing the activity of the HRV 5′UTR (Fig. 7g, h).

Our finding of the importance of CXCL8 in the replication process of various respiratory enteroviruses prompted us to further investigate the impact of this signaling pathway on other crucial human respiratory viruses. Influenza A virus (IAV) was then included in our study because of its differing replication cycle, which takes place in both the cytoplasm and the nucleus, unlike that of EV-D68 and HRV, which takes place solely within the cytoplasm. IAV infection also increased CXCL8 expression (Fig. 7i). IAV infection activated the MAPK pathway by stimulating CXCL8 signaling (Supplementary Fig. 13c, d). We further demonstrated that the IAV protein NS1 but not the IAV protein PB1-F2 effectively induced a dose-dependent increase in CXCL8 expression (Fig. 7j). Importantly, blocking CXCL8 strongly inhibited IAV replication (Fig. 7k). Similar to EV-D68 and HRV infection, IAV infection induced cytoplasmic accumulation of hnRNP-K, and its translocation was impaired by CXCL8 silencing (Fig. 7l). Recent studies have shown that hnRNP-K is a crucial viral RNA-binding protein for IAV, and mutant viruses lacking the ability to bind hnRNP-K exhibit significant defects in the nuclear export of M mRNA, resulting in the nuclear retention of viral RNA[26,41]. In our immunofluorescence system, we observed colocalization of M mRNA and the hnRNP-K protein in the cytoplasm of infected cells (Supplementary Fig. 14). Moreover, knockdown of endogenous hnRNP-K led to substantial nuclear retention of M mRNA during IAV infection (Supplementary Fig. 15), further supporting the involvement of hnRNP-K in the nuclear export of viral RNA. We employed two different methods, the nuclear-cytoplasmic fractionation assay followed by RT–PCR and RNA fluorescence in situ hybridization (FISH), to confirm that CXCL8 knockdown disrupts the cytoplasmic accumulation of hnRNP-K and impairs the nuclear export of M mRNA (Fig. 7m–p). Hence, IAV exploits the CXCL8-MAPK-hnRNP-K signaling axis to increase the nuclear export of viral mRNA.

Additionally, we employed a laboratory-established SARS-CoV-2 protein expression system to comprehensively investigate the

regulatory effects of coronaviral proteins on *CXCL8* promoter activity. Importantly, our findings demonstrated that both the structural protein E and the nonstructural protein ORF3a of SARS-CoV-2 effectively induced dose-dependent expression of CXCL8 (Supplementary Fig. 16a, b). These results imply that, in addition to the initial activation mediated by structural proteins during early viral infection stages, the viral accessory protein ORF3a continuously stimulates the expression and secretion of CXCL8 during later stages of progeny virion production. Notably, via the replicative SARS-CoV-2 infection reporter system, we provide compelling evidence that robust inhibition of CXCL8 significantly impedes efficient viral replication (Supplementary Fig. 16c). Furthermore, all the viral proteins described above rely on the expression of the endogenous syk protein to induce CXCL8 expression (Supplementary Fig. 17), suggesting that productive and effective signaling through CXCL8 is conserved among EV-D68, rhinoviruses, influenza viruses and SARS-CoV-2.

## Discussion
Owing to the ability of respiratory RNA viruses for rapid transmission and their mechanisms of mutation that allow the evasion of vaccine-induced immune neutralization, we are still in a passive position regarding the defence against respiratory RNA viruses as well as related drug research and development. However, humans are currently facing and will continue to face persistent public health threats from respiratory viruses; thus, the identification of conserved universal drug targets for different viral strains is urgently needed. In this study, we revealed that a CXCL8 signaling axis in respiratory cells drives proviral effects during infection with human viruses, highlighting the potential of this axis as a universal target for drugs combating respiratory virus infections.

In this study, we present a comprehensive pathway through which the respiratory virus EV-D68 modulates the microenvironment of host cells. Specifically, upon viral entry into the cell, exposure of the VP4 structural protein concealed within the virion triggers the activation of syk-binding motifs on VP4 proteins, thereby inducing CXCL8 transcription via the syk-PIK3CA-AKT signaling cascade. Consequently, the CXCL8 protein is synthesized and secreted extracellularly. The secreted CXCL8 protein can activate the CXCR1/2 receptors on both infected cells and neighbouring uninfected cells, leading to the activation of the MEK-ERK phosphorylation pathway. Through phosphorylation-mediated interactions in the cytoplasmic compartment, the hnRNP-K protein translocates from the nucleus to bind with viral RNA. This interaction sustains high activity levels of the 5′UTR of the viral genome and potentially primes hnRNP-K proteins in neighbouring uninfected cells for accelerated viral replication (Supplementary Fig. 18). Encouragingly, this CXCL8 signaling pathway plays a crucial role in facilitating the replication of several important human viruses, including rhinoviruses, influenza viruses and SARS-CoV-2, suggesting a potential target for broad-spectrum antagonism against respiratory virus infection.

Through millennia of evolution, hosts have developed a robust innate immune system, enabling rapid identification of foreign viruses through pattern recognition receptors (PRRs)[42]. The activation of PRRs triggers the secretion of cytokines and

                                                                    

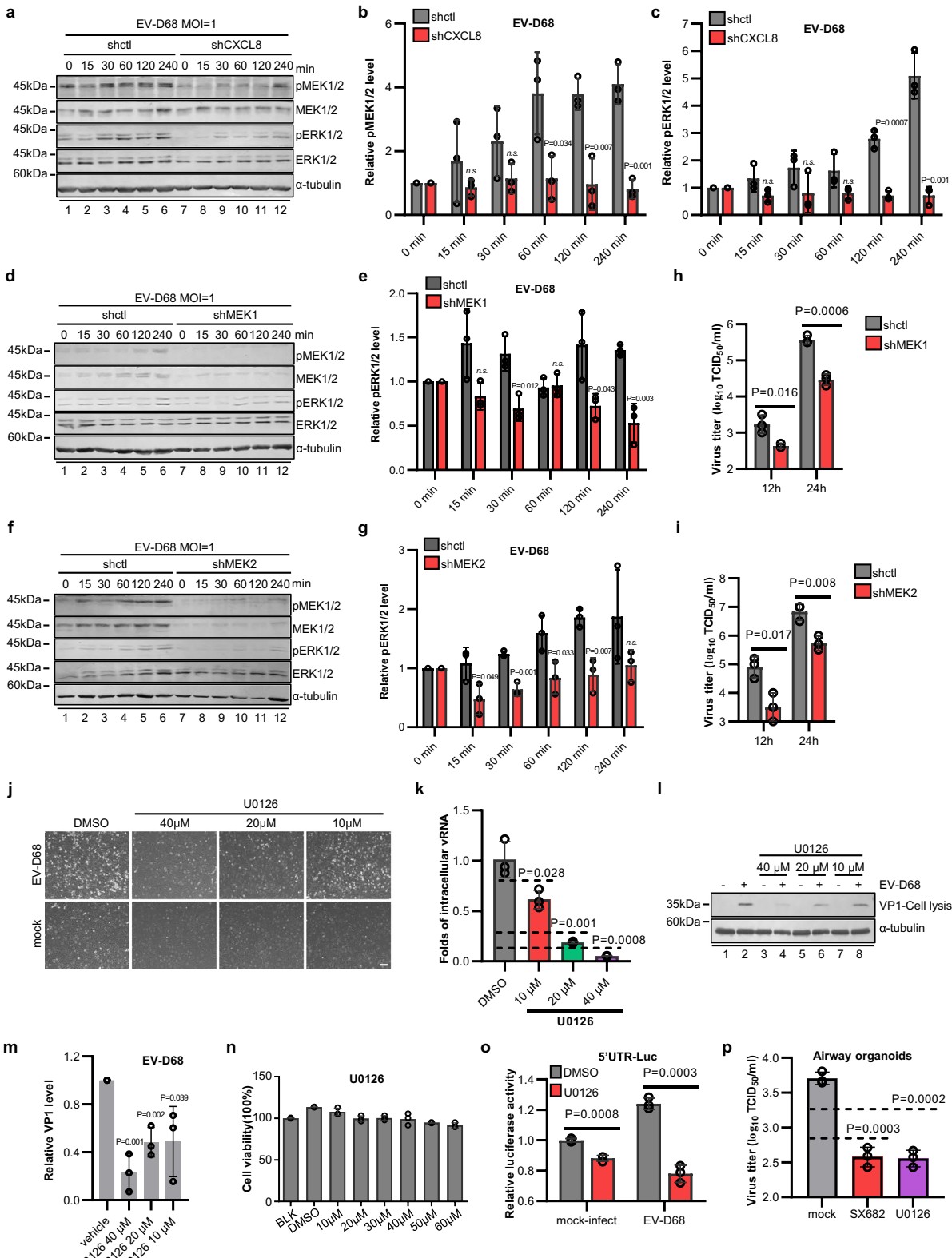

facilitates prompt transmission of danger signals associated with viral infection, inducing a state of increased combat readiness in neighbouring cells and recruiting immune cells to infected tissues for localized clearance[43,44]. Interestingly, viruses exploit a similar system to propagate signals that promote viral replication[43]. Infection with viruses, including rhinoviruses, influenza viruses and SARS-CoV-2, increases CXCL8 expression and secretion.

Through the release of CXCL8, infected cells not only establish a microenvironment conducive to viral propagation but also may render nearby uninfected cells more susceptible to infection.

The chemokine CXCL8 has comparable affinities for the GPCRs CXCR1 and CXCR2, which exhibit 76% sequence homology[45,46]. Both receptors participate in common GPCR-related pathways and regulate diverse cellular functions, including calcium release and activation of

**Fig. 4 | CXCL8 facilitates EV-D68 replication through the MAPK signaling pathway. a** EV-D68 infection activates MEK1/2-ERK1/2 signaling in a CXCL8-dependent manner. A549-shctl and A549-shCXCL8 cells were infected with EV-D68 (MOI = 1). The activation of the MEK-ERK signaling pathway was detected within 4 h post-infection. **b**, **c** Quantitative analysis of the relative pMEK1/2 or pERK1/2 protein levels in (**a**). **d**, **f** Silencing of either MEK1 (**d**) or MEK2 (**f**) inhibits the activating effect of EV-D68 on MAPK signaling. **e**, **g** Quantitative analysis of the relative pERK1/2 protein levels in (**d** and **f**). The value at 0 min post-infection was set as 1 for each group (**b**, **c**, **e**, **g**). **h**, **i** Silencing of MEK1 (**h**) or MEK2 (**i**) inhibits the production of progeny virions. **j**–**l** Treatment with the MEK1/2 inhibitor U0126 inhibits EV-D68 replication. **m** Quantitative analysis of the relative VP1 protein levels in (**l**). **n** Cytotoxicity assay with U0126. **o** U0126 treatment suppresses the function of the enteroviral 5′UTR. **p** Treatment with U0126 or SX682 inhibits EV-D68 replication in airway organoids. Human airway organoids differentiated for 28 days were infected with EV-D68 ($5.0 \times 10^5$ TCID$_{50}$) in the presence or absence of U0126 or SX682. The viral titer in the culture medium was measured 48 h post-infection. $N = 3$ (**b**, **c**, **e**, **g**–**i**, **k**, **m**–**p**) biological replicates. Representative of three biologically independent experiments (**a**, **d**, **f**, **j**, **l**). Data are represented as mean ± SD. Two-tailed $t$-test (**b**, **c**, **e**, **g**–**i**, **k**, **m**–**p**) was used to assess statistical significance. Scale bar in (**j**) is 100 μm. Source data are provided as a Source Data file.

the Ras-MAPK signaling pathway[30]. However, there is a disparity in the desensitization rate between these two receptors. While CXCR2 is rapidly internalized and inactivated at lower ligand concentrations, it returns to the cell surface significantly more slowly than CXCR1[47]. Moreover, CXCR2 can bind to more chemokine ligands than can CXCR1[48]. Recent studies have demonstrated that both receptors play crucial roles in mediating the CXCL8-induced proliferation, survival, migration, invasion, and angiogenic activity of endothelial cells; however, simultaneous knockdown of both CXCR1 and CXCR2 did not further increase the aforementioned physiological effects. These findings suggest that individual knockdown of either receptor is sufficient to modulate the angiogenic effect of CXCL8[49–51]. This finding is consistent with our results indicating that blockade of either CXCR1 or CXCR2 effectively inhibited the replication of EV-D68 induced by CXCL8.

The primary life cycle of many human respiratory RNA viruses occurs within the cytoplasm of host cells[52]. It is logical that, during viral infection, essential host elements required for processing, replication, and translation of viral RNA, which were originally localized in the nucleus, must be translocated into the cytoplasm to facilitate efficient viral replication[28]. This concept has been further supported by evidence demonstrating that specific viral proteins can effectively disrupt the nuclear transport pathways of intracellular proteins to impede the nuclear entry of newly synthesized proteins[52]. However, a more effective approach would involve expediting the translocation of preexisting cofactors for replication from the nucleus into the cytoplasm, thereby accelerating viral translation. Indeed, here, we elucidated the mechanism by which human respiratory viruses utilize host cytokine CXCL8 signaling to drive the cytoplasmic accumulation of nuclear hnRNP-K.

hnRNP-K, an RNA-binding protein, plays crucial roles in various cellular processes, including the regulation of gene expression, RNA splicing, and mRNA transport[53,54]. In the context of viral infection, hnRNP-K has been shown to modulate the replication capacity of specific viruses, such as enteroviruses[28], influenza viruses[26,27,55], and coronaviruses by interacting with specific elements within their genomes. The translocation of hnRNP-K into the cytoplasm increases its ability to recognize viral genomes. Notably, we observed a more potent effect of CXCL8 signaling on the replication of EV-D68 and HRV than on the activity of the 5′UTRs of these viruses. This discrepancy could be attributed to either a lack of sufficient sensitivity in our luciferase reporter system or to additional mechanisms of CXCL8 that promote viral replication separate from the increase in viral 5′UTR activity. In fact, while the CXCL8 signaling pathway facilitates the replication of all tested respiratory RNA viruses, the mechanism by which CXCL8 promotes viral replication varies across the viruses. Unlike the replication cycles of EV-D68 and HRV, which take place in the host cell cytoplasm, the replication cycle of IAV occurs partially in the nucleus. Previous studies have demonstrated that cellular hnRNP-K plays a crucial role as a viral RNA-binding protein for IAV, and our study further supports this finding by revealing that CXCL8 signaling promotes hnRNP-K translocation from the nucleus to the cytoplasm, facilitating the export of IAV M mRNA from the nucleus. Our mechanistic investigation revealed an intriguing phenomenon in which different RNA viruses exploit the native CXCL8-MAPK-hnRNP-K

axis in host cells to increase the activity of noncoding regions or aid in the nuclear export of viral mRNA. Whether there are other mechanisms by which CXCL8 promotes viral replication remains to be further explored. However, reversing virus-induced alterations in the subcellular localization of the hnRNP-K protein may become a direct and highly efficacious antiviral intervention strategy. Given the substantial dependence of viral replication on host hnRNP-K, human viruses may face formidable challenges in circumventing this impediment through sequence mutations.

The PI3K/AKT signaling pathway plays crucial roles in cell survival, apoptosis, proliferation and differentiation[56]. The PI3K/AKT signaling pathway is frequently hijacked by diverse viruses to facilitate viral entry, RNA replication or translation[57]; these viruses include hepatitis C virus[58], herpes viruses[59], and human immunodeficiency virus 1 (HIV-1)[60]. Here, we showed that EV-D68-mediated activation of the CXCL8 signaling cascade relies on a region mimicking the immunoreceptor tyrosine-based activation motif (ITAM) in the sequence of its structural protein VP4. The viral ITAM helps VP4 to recruit host syk proteins and activate the downstream PI3K/AKT pathway to increase CXCL8 gene expression. Interestingly, the ITAM mimic motif was found in several respiratory virus proteins. Whether this phenomenon is an evolutionary strategy learned from host cells during viral evolution is worthy of further study.

The cytokine CXCL8, also called interleukin-8, plays a pivotal role in orchestrating immune responses and promoting inflammation in various infectious and inflammatory conditions. Together with its receptors CXCR1 and CXCR2, CXCL8 significantly contributes to neutrophil chemotaxis and activation during inflammation, thereby reinforcing the first line of immune defence[61]. However, accumulating evidence has revealed that dysregulation of CXCL8 signaling is implicated in various inflammation-mediated respiratory diseases, such as cystic fibrosis, chronic obstructive pulmonary disease[62], and bronchial asthma[63]. CXCL8 also plays a role in the proliferation, invasion, and migration of lung cancer cells[64]. Hence, excessive production of CXCL8 is detrimental to human respiratory function. Several clinical studies have reported elevated levels of CXCL8 in patients infected with SARS-CoV-2[65] and influenza viruses[66]. Our findings indicate that, in addition to the direct cytotoxic effects of viral infection on the respiratory system, the demand for CXCL8 resulting from viral infection promotes the upregulation of CXCL8 beyond that required for an appropriate host immune response, thereby exacerbating pulmonary dysfunction. Empirical evidence from real-world data also suggests a robust correlation between the CXCL8 level in coronavirus disease 2019 (COVID-19) patients and the severity of illness, thereby providing additional support for this idea[67]. The clinical implication from this study for the guidance of future therapeutic interventions is that blocking CXCL8 signaling not only mitigates the excessive host inflammatory response caused by cytokines but also directly inhibits respiratory virus replication. This dual action effectively curtails both the sustained production of excessive inflammatory factors and the progression of symptoms, contributing to the overall amelioration of respiratory conditions. Additionally, it is crucial to consider the beneficial roles of CXCL8 in immune regulation, such as its ability to recruit neutrophils to inflammatory sites and its potential roles in the

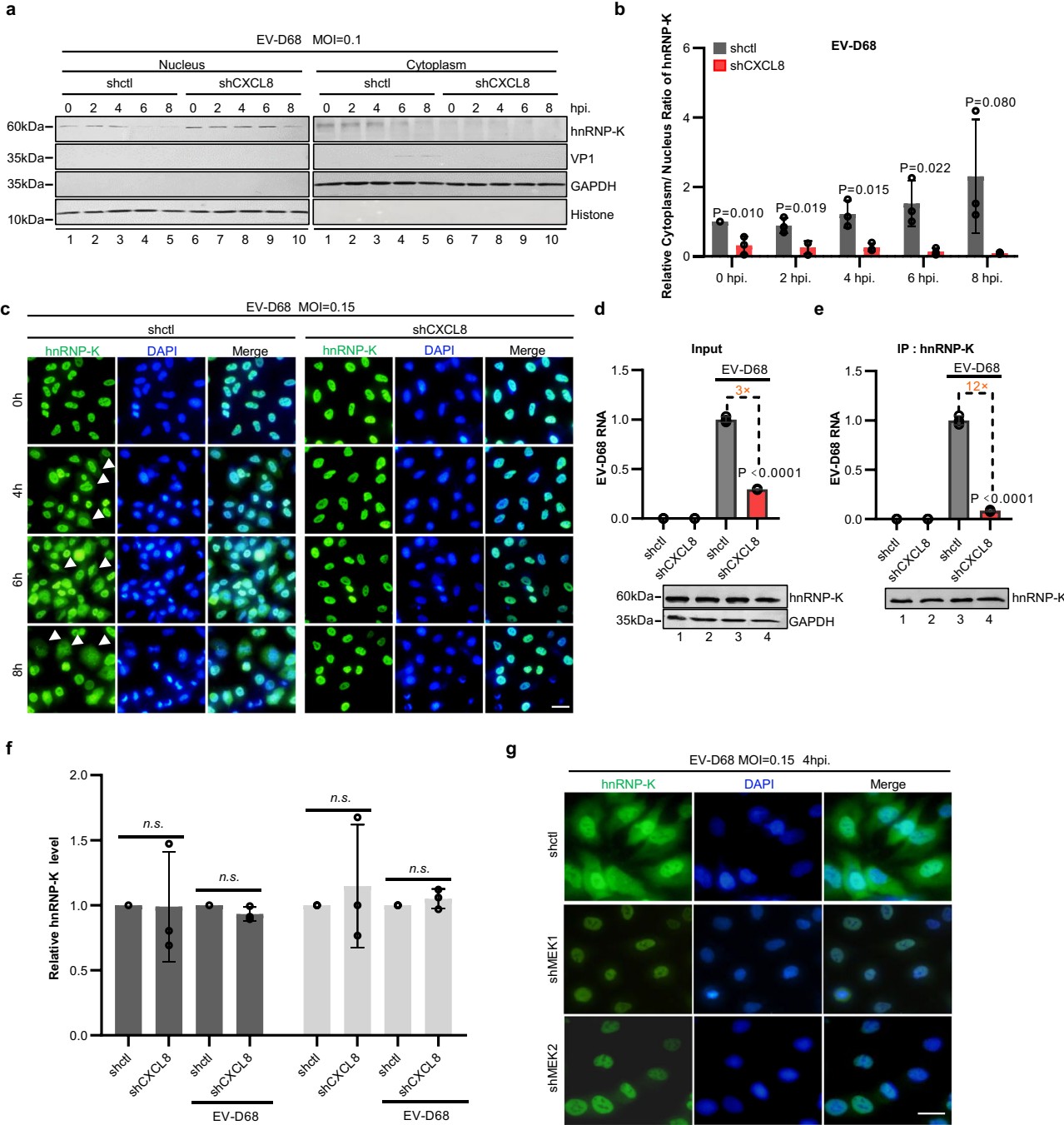

**Fig. 5 | EV-D68-CXCL8 signaling induces cytoplasmic accumulation of hnRNP-K. a** A549-shctl and A549-shCXCL8 cells were infected with EV-D68 (MOI = 0.1). Cell samples were collected within 8 h post-infection. The abundances of hnRNP-K and the viral protein VP1 in the cytoplasm and nucleus were assessed. **b** The relative cytoplasmic/nuclear ratio of hnRNP K in (**a**). **c** Immunofluorescence staining for hnRNP-K in A549-shctl and A549-shCXCL8 cells infected with EV-D68 (MOI = 0.15). **d**, **e** CXCL8 enhances the interaction of viral RNA with hnRNP-K. **f** Quantitative

analysis of the relative hnRNP K protein levels in (**d**, **e**). **g** Silencing of MEK1 or MEK2 blocks EV-D68-induced hnRNP-K translocation. $N = 3$ (**b**, **d**–**f**) biological replicates. Representative of three biologically independent experiments (**a**, **c**, **g**). Data are represented as mean ± SD. Two-tailed $t$-test (**b**, **d**–**f**) was used to assess statistical significance. Scale bar in (**c** and **g**) is 20 μm. Source data are provided as a Source Data file.

host defence against invading pathogens. Fortunately, we elucidated the signaling cascades through which respiratory virus infection activates CXCL8 signaling and CXCL8 promotes viral replication. By targeting MAPK or the cytoplasmic translocation of hnRNP-K, we can achieve a broad-spectrum antiviral effect similar to that of CXCL8 blockade while maintaining the other biological activities of CXCL8. However, one limitation of this study is its lack of in vivo experimental data, which is primarily because the genomes of the mice commonly

used for establishing viral infection models lack the CXCL8 coding sequence. To some extent, the data from our experiments in respiratory organoid models support the role of CXCL8 signaling in promoting viral replication. Future studies should focus on assessing the broad-spectrum antiviral effects of targeting CXCL8 in more realistic in vivo models.

Cellular cytokines function as mediators of intercellular communication, typically playing a role in transmitting danger signals

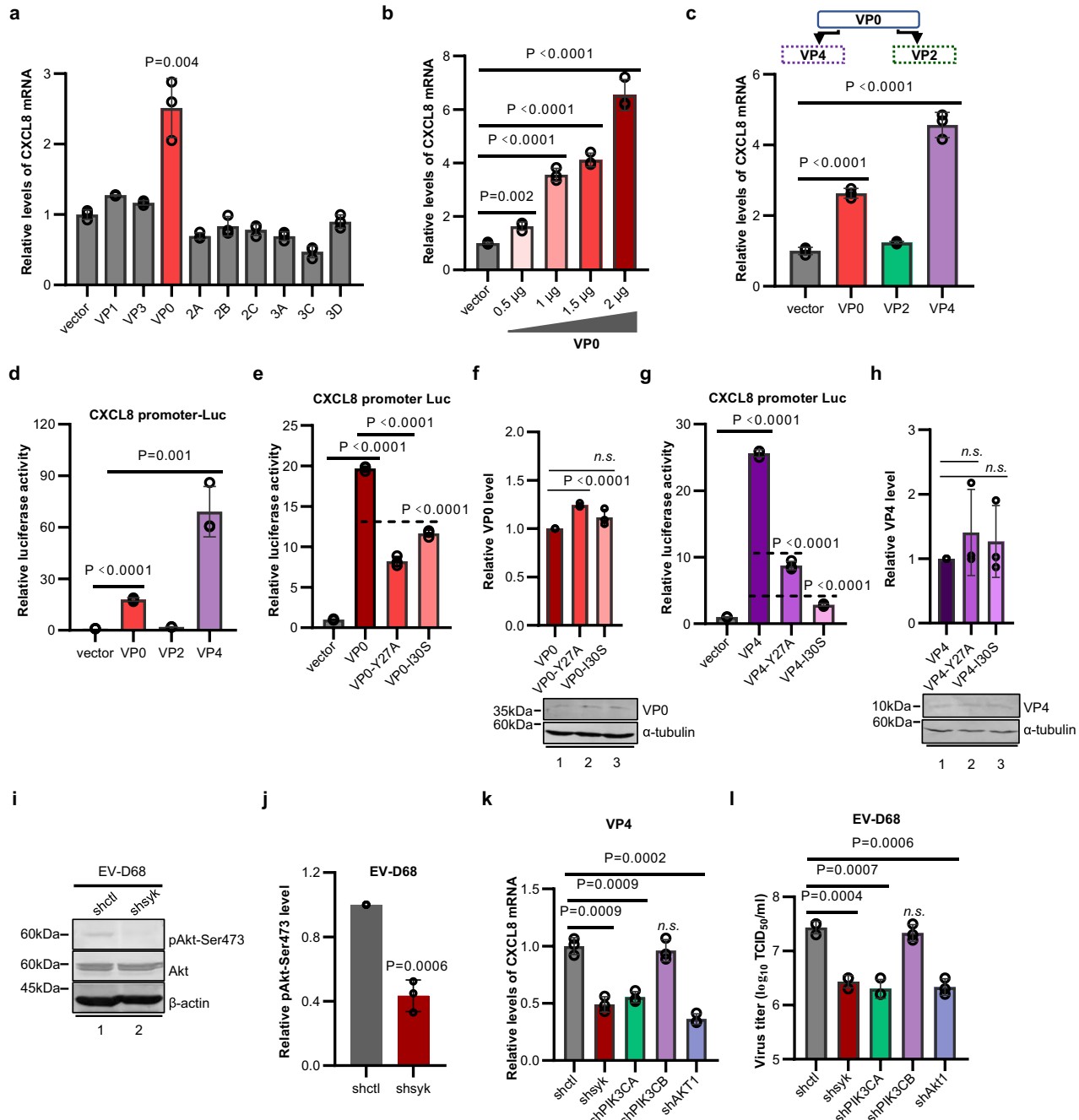

**Fig. 6 | VP4 activates CXCL8 via syk-PI3K signaling. a** VP0 induces CXCL8 expression. HEK293T cells were transfected with the vector or the expression plasmids for the viral proteins VP1, VP3, VP0, 2A, 2B, 2C, 3A, 3C, and 3D. The relative intracellular *CXCL8* mRNA level was measured via qRT–PCR 24 h after transfection. Cells transfected with vector group was set as control group. **b** Cells were transfected with increasing amounts of VP0. qRT–PCR analysis was performed 24 h after transfection to measure the intracellular *CXCL8* mRNA level. **c** VP4 induces CXCL8 expression. Schematic diagram of the cleavage of VP0 into VP4 and VP2. Cells were transfected with the vector or with the VP0, VP2 or VP4 expression plasmid. The relative intracellular *CXCL8* mRNA level was measured 24 h post-transfection. **d** VP4 increases the activity of the *CXCL8* promoter. HEK293T cells were co-transfected with *CXCL8*-promoter-Luc and the vector or the VP0, VP2 or VP4 expression plasmid. Luciferase activity was measured 24 h post-transfection. **e, g** Mutations in the "YXXI" motif in VP0 (**e**) and VP4 (**g**) interfere with the activation of CXCL8.

**f, h** Representative Western blot images depicting VP0 or VP4 expression levels in cells transfected with indicated plasmids, along with corresponding quantitative analysis of the relative VP0 or VP4 protein levels. **i** Silencing of syk inhibits EV-D68-induced Akt phosphorylation. **j** Quantitative analysis of the relative pAkt-Ser473 protein levels in (**i**). **k** VP4 induces CXCL8 expression through the syk-PIK3CA-Akt pathway. HEK293T-shctl, -shsyk, -shPIK3CA, -shPIK3CB and -shAkt1 cells were transfected with the VP4 expression plasmid. The relative *CXCL8* mRNA level was measured 24 h after transfection. **l** The syk-PI3KA-AKT axis is necessary for EV-D68 replication. A549-shctl, -shsyk, -shPIK3CA, -shPIK3CB, and -shAkt1 cells were infected with EV-D68 (MOI = 0.02) and the viral titers in the culture medium was measured at 24 hpi. *N* = 3 (**a–h**, **j–l**) biological replicates. Representative of three biologically independent experiments (**i**). Data are represented as mean ± SD. Two-tailed *t*-test (**a–h**, **j–l**) was used to assess statistical significance. Source data are provided as a Source Data file.

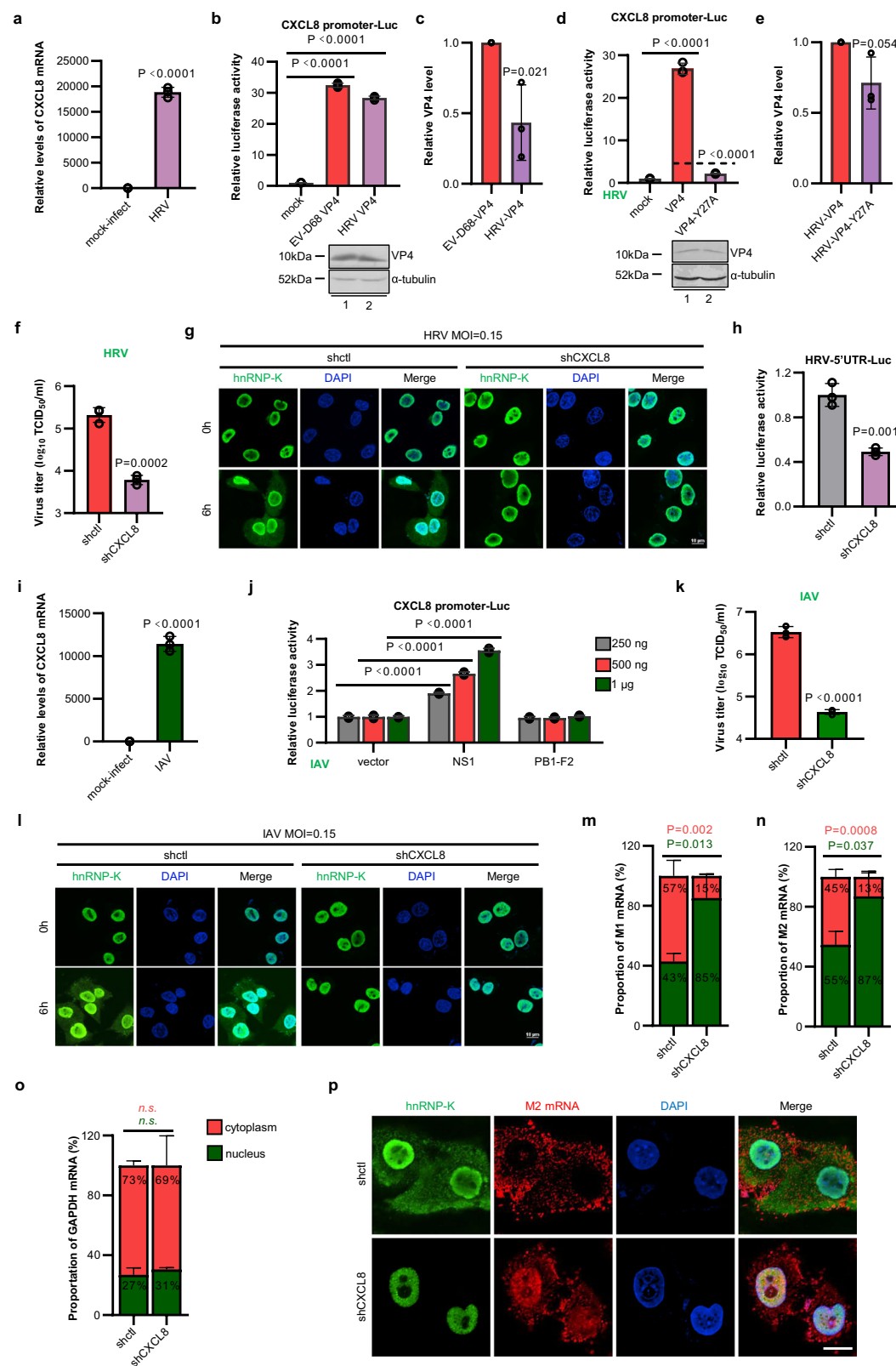

associated with pathogen invasion and prompting neighbouring cells to enter an antiviral state. Here, we discovered that respiratory virus infection specifically activates the expression and secretion of the cytokine CXCL8. Intriguingly, the virus exploits this host cell signaling pathway to promote viral replication, thereby transmitting signals to the surrounding microenvironment. This signaling cascade leads to the translocation of the nuclear cofactor hnRNP-K to the cytoplasm,

thereby increasing the recognition of viral RNA by this key cofactor. Our findings not only provide insights into a comprehensive cycle of viral manipulation of cells within the human respiratory system but also present a potential universal drug target within the affected signaling pathway for the development of repositories of candidate antiviral drugs, which is anticipated to aid endeavours to combat the current pandemic and future challenges posed by respiratory viruses.

**Fig. 7 | CXCL8 signaling is broadly required for human respiratory virus infection. a** The transcription of *CXCL8* was significantly increased upon HRV infection (MOI = 0.01, 48 hpi). **b** Similar to EV-D68 VP4, HRV VP4 activates the *CXCL8* promoter. **c** Quantitative analysis of the relative VP4 protein levels in (**b**). **d** Mutations in the "YXXL/I" motif of HRV VP4 impair its ability to activate CXCL8. **e** Quantitative analysis of the relative VP4 protein levels in (**d**). **f** Downregulation of CXCL8 interferes with the generation of HRV progeny virions. **g** Infection with HRV leads to the re-localization of hnRNP-K to the cytoplasm, which is dependent on the regulation of CXCL8. Immunofluorescence staining for hnRNP-K in A549-shctl and A549-shCXCL8 cells infected with HRV. **h** Knockdown of CXCL8 impaired the activation of the HRV-5′UTR. **i** IAV infection induces *CXCL8* transcription. **j** NS1 but not PB1-F2 of IAV facilitates *CXCL8* promoter activity in a dose-dependent manner. **k** Silencing of CXCL8 blocks the production of IAV progeny virions. **l** IAV infection induces cytoplasmic re-localization of hnRNP-K in a CXCL8-dependent manner. **m–o** Knockdown of CXCL8 results in the nuclear accumulation of IAV M1 and M2 mRNAs. A549-shctl and A549-shCXCL8 cells were infected with IAV at an MOI of 1, and the relative abundances of IAV M1 or M2 mRNA in the nucleus and cytoplasm were measured by qRT–PCR after cell fractionation at 8 h post-infection. **p** A549-shctl and A549-shCXCL8 cells were infected with IAV at an MOI of 0.15. RNA FISH and immunofluorescence staining were performed after 8 h to evaluate M2 mRNA and the hnRNP-K protein. *N* = 3 (**a–f**, **h–k**, **m–o**) biological replicates. Representative of three biologically independent experiments (**g**, **l**, **p**). Data are represented as mean ± SD. Two-tailed *t*-test (**a–f**, **h**, **i**, **k**, **m–o**) or two-way ANOVA (**j**) was used to assess statistical significance. Scale bar in (**g** and **l**) is 10 μm. Scale bar in (**p**) is 5 μm. Source data are provided as a Source Data file.

## Methods

### Plasmids and reagents

The expression vectors for EV-D68 2A, EV-D68 2B, EV-D68 2C, EV-D68 3A, EV-D68 3C, EV-D68 3D, pol I-HRV16-5′UTR-Luc-pHH21, HIV 1-LTR-Luc, HRV VP4, IAV NS1, IAV PB1-F2, PCBP1-HA, and PCBP2-HA were purchased from Generay Biotech Co., Ltd. (19711-19716, 27619, 3012, 20119-20123). EV-D68 VP0-G2A, EV-D68 VP0-Y27A, EV-D68 VP0-I30S, EV-D68 VP4-Y27A, EV-D68 VP4-I30S, and HRV VP4-Y27A were generated through site-specific mutagenesis. The expression vectors for SARS-CoV-2 proteins (NSP1, NSP2, NSP4, NSP5, NSP6, NSP7, NSP8, NSP9, NSP10, NSP11, NSP12, NSP13, NSP14, NSP15, ORF3a, ORF3b, ORF6, ORF7b, ORF8, ORF9b, ORF9c, ORF10, E, M, N) were gifts from Dr Nevan Krogan (Addgene, USA)[68]. The *CXCL8/IL8* promoter–luciferase construct (*CXCL8* promoter-Luc) was generated by inserting the 1.4 kb sequence of the *CXCL8/IL8* promoter into the pGL3-basic vector[69]. The pol I-EV68-5′UTR-Luc-pHH21, EV-D68 VP0, VP1, and VP2 expression plasmids were kindly provided by Dr Tao Wang (Tianjin University)[29]. The small interfering RNA (siRNA) against hnRNP-K was purchased from RiboBio (siBDM1999A).

The reagents used in this study were as follows: anti-EV-D68 VP1 antibody (Genetex, GTX132312); anti-β-actin antibody (Sigma, A3853); anti-α-tubulin antibody (GenScript, A01410); anti-glyceraldehyde-3-phosphate dehydrogenase (GAPDH) antibody (GenScript, A01622); anti-histone H3 antibody (Abcam, ab176842); anti-HA antibody (Thermo, 71-5500); anti-MEK1/2 antibody (Cell Signaling Technology, 8727); anti-pMEK1/2 antibody (Cell Signaling Technology, 9154); anti-ERK1/2 antibody (Cell Signaling Technology, 4695); anti-pERK1/2 antibody (Cell Signaling Technology, 4370); anti-hnRNP-K antibody (Cell Signaling Technology, 4675); anti-PI3K-p110α antibody (Cell Signaling Technology, 4249); anti-PI3K-p110β antibody (Cell Signaling Technology, 3011); anti-Akt antibody (Cell Signaling Technology, 4691); rCXCL8 (Abcam, ab259397); SX682 (MCE, HY-119339); Danirixin (MCE, HY-19768); C16-PAF (MCE, HY-108635) and U0126 (MCE, HY-12031A); Human IL-8/CXCL8 Antibody (R&D System, MAB208).

### Cells and viruses

Human lung adenocarcinoma A549 cells (CCL-185, ATCC), human bronchial epithelial BEAS-2B cells (CRL-9609, ATCC), human bronchial adenocarcinoma CALU-3 cells (HTB-55, ATCC), human rhabdomyo-sarcoma RD cells (CCL-136, ATCC), HEK293T cells (CRL-3216, ATCC), H1-HeLa cells (CRL-1958, ATCC), Madin–Darby canine kidney (MDCK) cells (GN023, Stem Cell Bank, Chinese Academy of Sciences) and Caco-2-N cells (provided by Dr Qiang Ding) were cultured in DMEM (Sigma–Aldrich, USA) supplemented with 10% FBS (Biological Industries, USA) and 1% penicillin/streptomycin. Primary HBECs, isolated from non-smoking healthy donors and commercially available from ScienCell (3210), were cultured in airway epithelial cell basal medium (ATCC PCS-300-030) supplemented with the components of a bronchial epithelial cell growth kit (ATCC PCS-300-040).

The EV-D68 prototype strain Fermon (VR-1826, ATCC) and circulating strains US/MO/14-18947 (VR-1823D, ATCC) and US/KY/14-18953 (VR-1825D, ATCC) were propagated in RD cells and stored at −80 °C. The cloned full-length complementary DNA (cDNA) of RV-A16 (pR16.11, Cat. No. VRMC-8) was obtained from ATCC. The recovered rhinoviruses were propagated in H1-HeLa cells. The human influenza virus strain A/PR/8/34 (VR-1469, ATCC) was propagated in MDCK cells. SARS-CoV-2 GFP/ΔN was a gift from Dr Qiang Ding (Tsinghua University).

### Establishment and differentiation of airway organoids

Human airway organoids were generated with the PneumaCult Airway Organoid Kit (05060, STEMCELL Technologies) according to the manufacturer's instructions. In brief, HBECs were seeded in a Matrigel dome and cultured for one week in PneumaCult Airway Organoid Seeding Medium. The medium was changed every 2 days by carefully aspirating the medium and adding 750 μl of Airway Organoid Seeding Medium. The organoids were subsequently cultured for four weeks in PneumaCult Airway Organoid Differentiation Medium.

### Lentivirus production and gene silencing

HEK293T cells were cotransfected with the following plasmids: shRNA plus pRSV-Rev (12253, Addgene), pMDLg/pRRE (12251, Addgene), and pCMV-VSV-G (8454, Addgene). The lentivirus-containing supernatant was harvested 48 h post-transfection, and lentiviral particles were purified via centrifugation. The concentrated virus was then used to transduce the cells of interest. Puromycin (1.5 μg/ml) was added to the culture medium after 2 days of transduction. The knockdown efficiency of each target gene was examined by immunoblotting or quantitative reverse transcription PCR (qRT–PCR).

The targeting sequences of each shRNA were as follows: shCXCL8, 5′-GAAGCGCTACTTGGTCAAATT-3′; shCXCR1, 5′-GAAGCGCTACTTGGT-CAAATT-3′; shCXCR2, 5-GAAGCGCTACTTGGTCAAATT-3′; shMEK1, 5′-GCTTCTATGGTGCGTTCTACA-3′; shMEK2, 5′-CTTCCAGGAGTTTGT-CAATAA-3′; shsyk, 5′-GCAGATGGTTTGTTAAGAGTT-3′; shPIK3CA, 5′-GCATTAGAATTTACAGCAAGA-3′; shPIK3CB, 5′-CGACAAGACTGCCGA-GAGATT-3′; and shAkt1, 5′-CGCGTGACCATGAACGAGTTT-3′.

### Cell viability assay

A total of $1.5 \times 10^4$ A549 cells per well were seeded into a 96-well plate in culture medium supplemented with DMSO or with a concentration gradient of the test compound and incubated for 24–48 h. Then, the medium was replaced, and 10 μl of Cell Counting Kit-8 (CCK-8) reagent (MCE, Cat. No. HY-K0301) was added to each well. The plate was incubated at 37 °C for 1–3 h, and the absorbance was then measured at 450 nm.

### Quantitative reverse transcription PCR (qRT-PCR)

Total RNA was extracted from cells or supernatants with TRIzol reagent (Life Technologies). The RNA was then transcribed into cDNA with a reverse transcription kit (TransGen Biotech, China). The cDNA was then used as the template for PCR amplification, which was carried out on a LightCycler 480 Instrument II (Roche480, Switzerland) with SYBR Green (Monad Biotech Co., Ltd, China). The thermal cycling conditions used

for PCR were as follows: 95 °C for 5 min, followed by 40 cycles at 95 °C for 10 s, 60 °C for 20 s and 72 °C for 20 s. qRT–PCR was performed using the following primer sequences: GAPDH forward primer, 5′-GCAAATTCCATGGCACCGT-3′; GAPDH reverse primer, 5′-TCGCCCCACTTGATTTTGG-3′; EV-D68 forward primer, 5′-TGTTCCC ACGGTTGAAAACAA-3′; EV-D68 reverse primer, 5′-TGTCTAGCGTCTC ATGGTTTTCAC-3′; CXCL8 forward primer 5′-TTCAGAACTTC GATGCCAGT-3′; CXCL8 reverse primer 5′-GGGCCACTGTCAATCACT CT-3′; spleen tyrosine kinase (syk) forward primer, 5′-CATGGA AAAATCTCTCGGGAAGA-3′; syk reverse primer, 5′-GTCGATGCGATA GTGCAGCA-3′; PI3K-p110α forward primer, 5′CACC-3′; PI3K-p110α reverse primer, 5′-TTTCGCACCACCTCAATAAG-3′; IAV M1 mRNA forward primer, 5′-ATCAGACATGAGAACAGAATGG-3′; IAV M1 mRNA reverse primer, 5′-TGCCTAGCCTGACTAGCAACCTC-3′; IAV M2 mRNA forward primer, 5′- GAAAATTTGCAGGCCTATCAGAAAC-3′; and IAV M2 mRNA reverse primer, 5′- CCAATGATATTTGCGGCAATAGCGAG-3′. The relative levels of the target RNAs were calculated via the comparative 2-ΔΔCT method with normalization to the GAPDH mRNA level.

## Viral attachment assay

Cells were incubated with EV-D68 (MOI = 1) at 4 °C or 37 °C for 2 h and were then washed with phosphate-buffered saline (PBS) to remove unbound virus. Total RNA was extracted, and the relative viral RNA abundance was measured via qRT–PCR.

## Virus titer assay

Viral titers were determined via an endpoint dilution assay (EPDA). In brief, RD cells were preseeded in 96-well plates at a density of $1 \times 10^4$ cells/well. The virus sample was serially diluted (tenfold) in DMEM containing 1% FBS and added to the wells. Viral titers were determined by evaluating the appearance of CPEs via a microtiter assay in accordance with the Reed–Muench method.

## Immunoblotting

Cell or supernatant samples were collected and boiled in 1× loading buffer (0.08 M Tris, pH 6.8; with 2.0% sodium dodecyl sulfate [SDS], 10% glycerol, 0.1 M dithiothreitol [DTT], and 0.2% bromophenol blue), and proteins were separated on an SDS–polyacrylamide gel electrophoresis (PAGE) gel and transferred to a nitrocellulose (NC) membrane with a semidry transfer apparatus (Bio–Rad, USA). The membrane was incubated first with primary antibodies overnight at 4 °C and then with the corresponding alkaline phosphatase-conjugated secondary antibodies (Jackson ImmunoResearch Laboratories). The membrane was stained with 5-bromo-4-chloro-3-indolyl phosphate (BCIP) and nitroblue tetrazolium chloride (NBT) (Sigma–Aldrich) and visualized for quantification of band densities. Uncropped immunoblots are depicted in the Source Data File.

## Enzyme-linked immunosorbent assay (ELISA)

CXCL8 in the cell culture medium was quantified with an ELISA kit (Multisciences EK108-03, China) according to the manufacturer's instructions. The bottom surface of a 96-well ELISA microplate was precoated with a capture antibody specific for CXCL8. CXCL8 standards and samples containing CXCL8 were pipetted in triplicate into these wells. A biotinylated detection antibody that recognizes CXCL8 was then added. During the first incubation step, the standards or samples and the detection antibody were simultaneously incubated. After washing, the enzyme streptavidin-HRP was added, and the plate was incubated and washed. 3,3′,5,5′-Tetramethylbenzidine (TMB) substrate solution was added, which was acted upon by the enzyme conjugate to generate a coloured reaction product. The intensity of this coloured product was directly proportional to the concentration of CXCL8 in each sample. The absorbance was measured at a wavelength of 450 nm with a BioTek ELISA Reader (BioTek Instruments, Inc., USA).

## Nuclear–cytoplasmic fractionation

All preparation steps were performed on ice. Cells were collected and washed with PBS twice. The cells were resuspended in 70 μl of lysis buffer (10 mM HEPES-NaOH, pH 7.9; 10 mM KCl; 1.5 mM $MgCl_2$; and 0.5 mM beta-mercaptoethanol supplemented with a protease inhibitor) and incubated on an end-over-end rotator for 20 min at 4 °C. Five microlitres of 10% NP-40 was added to this solution, which was subsequently vortexed, incubated on ice for 2 min, and centrifuged at $16,000 \times g$ for 15 min. The supernatant (cytoplasmic fraction) was collected and boiled in 1× loading buffer. The pellet (nuclear fraction) was washed with PBS twice and then boiled in 1× loading buffer. The proteins in the nuclear and cytoplasmic fractions were analysed by immunoblotting. Subcellular fractionation of IAV M1 mRNA, M2 mRNA and GAPDH mRNA was conducted with a PARIS™ kit (AM1921, Thermo), and qRT–PCR analysis was then performed.

## RNA binding protein immunoprecipitation (RIP)

Protein G beads (Invitrogen) were added to the anti-hnRNP-K antibody and incubated for 1 h at 4 °C with gentle rotation. The beads were then washed and resuspended in RIP buffer. HEK293T-shctl and HEK293T-shCXCL8 cells were infected with EV-D68 (MOI = 0.15) or mock-infected for 12 h. The cells were then harvested and lysed in 1 ml of ice-cold lysis buffer (150 mM Tris, [pH 7.5; 150 mM NaCl; 1% Triton X-100; and protease inhibitor) by incubation with rotation for 1 h. The lysate was then centrifuged at $10,000 \times g$ for 30 min. A portion of the cell lysate was saved as Input (both EV-D68 RNA and hnRNP-K Input), and the other part was subjected to precipitation with hnRNP-K–protein G beads at 4 °C for 3 h or overnight. The beads were subsequently washed eight times with wash buffer (20 mM Tris, pH 7.5; 100 mM NaCl; 0.1 mM EDTA; and 0.05% Tween 20). The coprecipitated RNA was isolated by the TRIzol method and analysed via qRT–PCR. Proteins isolated by the beads were eluted with elution buffer (0.1 M glycine–HCl, pH 2.0) and analysed by immunoblotting.

## RNA FISH

Cells on slides were fixed for 30 min with 4% paraformaldehyde (PFA) and were then permeabilized for 20 min with 0.5% Triton X-100. Next, the prehybridization solution was added dropwise to the slides, which were then incubated in a 40 °C incubator for 1 h. The prehybridization solution was discarded, the probe-containing hybridization solution was added, and the slides were incubated with mixing overnight in a heated chamber. For the IAV M2 mRNA FISH experiments, DNA probes (M2 probe sequence, tctgataggcgtttcgacct) and RNASwe AMI™ with Cy3 Dye Kit (GF002-50T) purchased from Servicebio were used. The following day, the hybridization solution was removed by washing with serial dilutions of saline sodium citrate (SSC) buffer. Then, the corresponding branch probe hybridization solution was added, and the slides were placed horizontally in a wet box at 40 °C for 45 min. The slides were rinsed with SSC buffer again, and the signal probe was added for a 3-h incubation at 40 °C. After a final wash with SSC buffer, the samples were blocked with BSA and then subjected to immunofluorescence staining; alternatively, the nuclei were stained with 4′,6-diamidino-2-phenylindole (DAPI), and images of the slides were acquired with a Nikon upright fluorescence microscope.

## Immunofluorescence assay

Cells were fixed with 4% PFA for 30 min, permeabilized with 0.1% Triton X-100 for 10 min, and blocked in 5% BSA solution for 1 h. Then, the cells were incubated first with an anti-hnRNP-K antibody overnight at 4 °C and then with an Alexa Fluor 594-conjugated goat anti-rabbit secondary antibody (Life Technologies, A-11012) for 1 h at room temperature. The nuclei were stained with DAPI. Fluorescence images were acquired with a fluorescence microscope (Olympus).

## Statistics and reproducibility

All the statistical analyses were performed with GraphPad Prism software (version 8.0) via two-tailed $t$-test or two-way ANOVA. Data were shown as the mean ± SD. Statistical significance was assumed at $P < 0.05$. No statistical method was used to predetermine sample size. No data were excluded from the analyses. The experiments were not randomized. The Investigators were not blinded to allocation during experiments and outcome assessment.

## Reporting summary

Further information on research design is available in the Nature Portfolio Reporting Summary linked to this article.

## Data availability

The experimental data generated in this study are provided in the main text, figures, supplementary information or source data file and are available within the article. The RNA-seq data generated in this study have been deposited in the GEO database under accession code GSE273371. Source data are provided with this paper.

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

## Acknowledgements

We thank Ling Xue and Yuanyuan Li for technical assistance. We thank Drs. Q. Ding, and T. Wang provided the critical reagents. This work was supported by the NSFC Excellent Young Scientist Fund (32222005 to W.W.), the National Natural Science Foundation of China (82372226 to W.W., 82172246 to H.G.), the National Major Project for Infectious Disease Control and Prevention (2018ZX10731-101-001-016), the Department of Science and Technology of Jilin Province (No. 20210101015JC to H.G.), the Open Project of Key Laboratory of Organ Regeneration and Transplantation, Ministry of Education, the Program for JLU Science and Technology Innovative Research Team (2017TD-08), Fundamental Research Funds for the Central Universities, and the Young Scientist Development Fund of the First Hospital of Jilin University (JDYY15202412 to Q.Y.).

## Author contributions

W.W. conceived and designed the experiments; Q.Y., H.G., H.L., and Z.L., participated in multiple experiments; W.W., Q.Y., F.N., Z.W., K.L., and H.K., analyzed the data; W.W. and Q.Y., wrote the manuscript with help from all authors.

## Competing interests

The authors declare no competing interests.
