## [Transparent Peer Review file · Nature Communications]

A CXCL8 signaling axis enables susceptibility to infection by respiratory enterovirus D68

Corresponding Author: Dr Wei Wei

Version 0:

Reviewer comments:

Reviewer #1

(Remarks to the Author)
Comments

In this study by Yang et. al, the authors provide a detailed exploration of the interplay between the respiratory enterovirus EV-D68 and the proviral chemokine CXCL8 in facilitating viral replication. The paper delves into the mechanisms by which EV-D68 exploit the CXCL8-CXCR1/2-MEK/ERK-hnRNPk signalling axis to promote virus replication, highlighting the potential of CXCL8-related signalling cascades as therapeutic targets for combating respiratory viral infections. The authors also demonstrate the requirement of CXCL8 for IAV, SARS-CoV-2 and HRV virus replication. Overall, the manuscript is well-organized and effectively structured, guiding the reader through the complex subject matter in a logical manner. Nevertheless, some points remain to be addressed experimentally to support the claims.

1. The effect of IAV and HRV infection on CXCL8 expression, MEK/ERK phosphorylation and eventually on hnRNPk re-localization to cytoplasm needs to be checked. This is essential to claim the broad-spectrum nature of this signalling axis for respiratory viruses.
2. As per the proposed signalling axis, treatment of cells with CXCL8 should lead to HNRNPk re-localization to the cytoplasm. Should be tested experimentally.
3. The role of this CXCL8 in cytokine storm and severe disease is plausible; however, this manuscript lacks data related to this assumption. Some in vivo virus infection data, where inhibitors of CXCL8 signalling have such impact would be necessary to make these claims.

Other minor comments are as follows

Figure 2j: Include experimental details for quantifying 5' UTR activity. Please provide a brief write-up and add relevant references.

Figure 2i, 4k: Details of infection conditions need to be mentioned in the legend and Methods section as well.

Lines 193-196: No results are shown for screening and validation of CXCL8 interacting partners. Proteomics Data and selection criteria for choosing hnRNPk should be provided.

Lines 226-229: Methodology for the establishment of a luciferase detection system based on CXCL8 promoter activity is not provided. Authors should update the methods section for all relevant experiments mentioned in the results section.

I suggest a grammar check for the entire manuscript. Some of the errors are listed

- Line 57 – correct word 'incubators'
- Line 105: Avoid 'Remarkably', 'dramatically'
- Line 119: 'Eliminate' incorrect use
- Line 124: 'final' should be finally
- Line 219 – 'Unbiased', remove 'an'.

Reviewer #2

(Remarks to the Author)

Reviewer #3

(Remarks to the Author)

Yang et al show that infection of A549 cells with EV-D68 induces expression and secretion of the proinflammatory cytokine CXCL8 or IL8. Silencing CXCL8 in multiple cell lines and in primary cells reduced EV-D68 replication, suggesting CXCL8 has a proviral effect. Replication was restored by exogenously added CXCL8. Targeting the CXCL8 signaling pathway, by downregulation of the receptor, or by known inhibitors, also inhibit viral replication. The authors attribute the proviral activity of CXCL8 to the activation of the MEK/ERK pathway, leading to hnRNPK translocation to the cytoplasm, that promotes translation from the 5' UTR of the viral RNA. The authors also show that VP4 of EV-D68 is responsible for the CXCL8 upregulation through PI3K activation. Finally, they show that CXCL8 is also induced by other viruses, such as influenza and SARS-CoV-2, and it is also proviral. While the impact of CXCL8 in vitro in viral replication is clear from the data, the mechanistic data are not conclusive, and the biological impact in vivo remains unknown.

Specific comments

1. Fig 2j: Specificity of the inhibition of the 5'UTR by silencing CXCL8 should be shown by using an irrelevant 5'UTR. In addition, the impact of inhibitors in this assay was quite minimal, and perhaps CXCL8 signaling exerts its proviral activity at a different level from that of impacting 5' viral UTR activity.
2. The experiments to define the pathways activated by CXCL8 that promote viral replication are not easily interpretable. First, the inability of the virus to induce MEK1/2-ERK1/2 signaling in a CXCL8 dependent manner might simply reflect that viral replication (inhibited by silencing CXCL8) is required for MEK1/2-ERK1/2 signaling. Does addition of CXCL8 to the cells in the absence of viral replication activate MEK1/2-ERK1/2 signaling? Same applies to translocation of hnRNPK. Does addition of CXCL8 to the cells in the absence of viral replication results in cytoplasmic translocation of hnRNPK, and is this dependent on MEK/ERK signaling? The authors would need to somehow restore viral replication in the absence of CXCL8 by activating the proposed downstream activities of CXCL8 in a CXCL8 independent manner (same way as they demonstrated that exogenous CXCL8 treatment compensates for silencing of CXCL8).
3. While it is quite interesting that other viruses activate CXCL8 expression and that CXCL8 is also proviral, this raises concerns on whether the mechanism on how CXCL8 enhances viral replication is actually the one proposed by the authors, as the two other viruses do not share the same 5'UTR of EV-D68. Also, as in vivo data are lacking on the impact of CXCL8 on viral replication and clearance, it is not a given that CXCL8 is proviral in vivo. Perhaps CXCL8 is required for the recruitment to the infected tissue of immune antiviral cells, and that its proviral impact is shadowed by being required for a proper antiviral immune response in vivo by attracting neutrophils to the site of infection.

Reviewer #4

(Remarks to the Author)

NCOMMS-24-14080-T review

The manuscript describes an interesting enhancement of EV-D68 replication when signaling through the CXCL8 pathway is induced during virus infection. CXCL8 signaling increases the cytoplasmic amounts of hnRNP K which binds to EV-D68 viral RNA and enhances viral gene expression. The EV-D69 capsid protein VP4 is also shown to mimic the function of ITAM and binds to syk kinase to increase PI3K/AKT signaling which results in increased production of CXCL8. Eliminating the interacting domain also eliminated VP4 activation of this pathway. Rhinovirus VP4 encodes a similar interaction domain and its activation of CXCL8 signaling was also dependent on the ITAM motif. Overall, the experiments are extensive and well controlled. This is an important observation for picornavirus family viruses.

Major points

Lines 193-196: This data is not shown or validated, which is really not appropriate because the results very much dictate the rationale for going after hnRNPK as a target.

I'm a little less enthusiastic about the data in figures 7 d-h. This data shows some dependence of other respiratory viruses on CXCL8 signaling, but the molecular basis for that requirement is less clear. Is hnRNP also involved here or is an entirely different mechanism mediating the CXCL8 effect here? This data seems a bit too preliminary. In lines 354-356 the authors continue to discuss mechanisms by which CXCL8 signaling might be aiding virus replication but influenza RNA replication occurs in the nucleus so it's difficult to see how the same mechanism mediated by hnRNP could positively affect a cytoplasmic and a nuclear replicating virus.

Minor points

Lines 57-63: this paragraph seems a bit odd. Adaptation usually doesn't occur quickly after infection and the first sentence seems more like literary writing as opposed to scientific writing.

Lines 86-87: What does "excessive background noise" mean?

Figure 2: the Figure 2C comparison is stated as being to uninfected cells but the figure appears to compare infected cells with a control or CXCR8 specific siRNA.

Figures 2i, 3c, 4d, 4f, 4k, 7c: is the y-axis in these figures in log₁₀ scale? If so, please label it as such.

Line 188: viruses don't think so they can't intentionally do anything.

Line 201: this is not real time monitoring, as the cells were fixed and processed for immunofluorescence analysis

Figure 1 legend: what are the comparison groups that form the basis of the statistical analysis? for example, is C being compared to mock infected or to 0h infection?

Figure 6i legend: indicate the MOI of infection the EV-D68 infections were performed at. This seems like a rather strong effect to be mediated by virions unless the infections were at an extremely high MOI.

Version 1:

Reviewer comments:

Reviewer #1

(Remarks to the Author)

The authors have addressed most of the comments to satisfaction through new experiments and editing the manuscript. The authors have explained that in vivo animal data is difficult for EV-68. However, it is not difficult for IAV. Nevertheless, they do provide EV-68 data in the Organoid infection model, which is an acceptable alternative.

Reviewer #2

(Remarks to the Author)

I co-reviewed this manuscript with one of the reviewers who provided the listed reports. This is part of the Nature Communications initiative to facilitate training in peer review and to provide appropriate recognition for Early Career Researchers who co-review manuscripts

Reviewer #3

(Remarks to the Author)

The authors have responded adequately to the issues pointed by this reviewer

Reviewer #4

(Remarks to the Author)

NCOMMS-24-14080A

The authors have responded appropriately to all the previous critiques and the manuscript has been strengthened significantly.

**Enclosed is a revised version of our manuscript “A CXCL8 Signaling Axis**
**Enables Susceptibility to Infection by Human Respiratory Viruses”. We**
**addressed each of the reviewers’ points in the letter below and made relevant**
**changes in the manuscript.**

**We are very grateful for the high quality reviews. It is obvious that each**
**reviewer read the manuscript very carefully. We truly appreciate the**
**feedback, which will substantially improve the reproducibility, quality and**
**clarity of our paper.**

**REVIEWER COMMENTS**

**Reviewer #1 (Remarks to the Author):**

**Comments**

In this study by Yang et. al, the authors provide a detailed exploration of the
interplay between the respiratory enterovirus EV-D68 and the proviral chemokine
CXCL8 in facilitating viral replication. The paper delves into the mechanisms by
which EV-D68 exploit the CXCL8-CXCR1/2-MEK/ERK-hnRNPK signalling axis to
promote virus replication, highlighting the potential of CXCL8-related signalling
cascades as therapeutic targets for combating respiratory viral infections. The
authors also demonstrate the requirement of CXCL8 for IAV, SARS-CoV-2 and
HRV virus replication.

Overall, the manuscript is well-organized and effectively structured, guiding the
reader through the complex subject matter in a logical manner. Nevertheless,
some points remain to be addressed experimentally to support the claims.

**Response:** We greatly appreciate the recognition of the potential importance of
our discoveries by the reviewer and find the comments both insightful and helpful.
Below are our point-by-point responses to the reviewer’s comments.

1. The effect of IAV and HRV infection on CXCL8 expression, MEK/ERK
phosphorylation and eventually on hnRNPK re-localization to cytoplasm needs to
be checked. This is essential to claim the broad-spectrum nature of this signalling
axis for respiratory viruses.

**Response:** Thank you very much for your suggestion. We have conducted all the
recommended experiments. Our data indicated that infection with either IAV or
HRV could increase the expression level of CXCL8 (new Figure 7a and 7g),
activate MAPK pathways (new Figure S14) and facilitate the cytoplasmic
translocation of the hnRNP K protein (new Figure 7e and 7j).

2. As per the proposed signalling axis, treatment of cells with CXCL8 should lead
to HNRNPK re-localization to the cytoplasm. Should be tested experimentally.

**Response:** We thank the reviewer for this important comment. We have
conducted the recommended experiments and confirmed that CXCL8 treatment
can sufficiently induce the relocalization of hnRNP K to the cytoplasm (new Figure
S11c).

3. The role of this CXCL8 in cytokine storm and severe disease is plausible;
however, this manuscript lacks data related to this assumption. Some in vivo virus
infection data, where inhibitors of CXCL8 signalling have such impact would be
necessary to make these claims.

**Response:** Thank you very much for pointing this out. Although mice are
frequently used as infection models in EV-D68 studies, they are not natural hosts
of EV-D68, and the mouse genome does not contain the CXCL8 coding sequence.
This limitation restricts our ability to conduct further in vivo infection studies. To
address this issue, we performed experiments using respiratory system organoids
and employed a CXCL8 neutralizing antibody and inhibitors of the downstream
factor MAPK, demonstrating effective suppression of EV-D68 replication through
the inhibition of CXCL8 signalling. Furthermore, we have incorporated a statement
acknowledging the limitations of our study into the revised manuscript (new Figure
2j and 4k).

Other minor comments are as follows

Figure 2j: Include experimental details for quantifying 5' UTR activity. Please
provide a brief write-up and add relevant references.

**Response:** We have incorporated related details on the 5'UTR reporter activity
assay into the revised manuscript (page 5, lines 126-130).

Figure 2i, 4k: Details of infection conditions need to be mentioned in the legend
and Methods section as well.

**Response:** We have added the recommended information to the revised
manuscript (page 15, lines 508-511, 520-526; page 28, lines 1041-1043; page 30,
lines 1072-1074).

Lines 193-196: No results are shown for screening and validation of CXCL8
interacting partners. Proteomics Data and selection criteria for choosing hnRNPK
should be provided.

**Response:** In this study, we observed that the expression of CXCL8 promoted EV-
D68 UTR activity. Consequently, we conducted a screen of host factors known to
significantly regulate EV-D68 UTR activity (PCBP1, PCBP2, and hnRNP K).
Interestingly, while the expression level of CXCL8 did not affect the ability of
PCBP1 and PCBP2 to bind to viral RNA (vRNA; new Figure S11a and S11b),
knockdown of CXCL8 decreased the ability of EV-D68 to induce the cytoplasmic
accumulation of hnRNP K (Figure 5b) and hindered the interaction of hnRNP K
with vRNA (Figure 5c and 5d). The relevant data have been incorporated into the
revised manuscript (page 7, lines 210-214).

Lines 226-229: Methodology for the establishment of a luciferase detection system
based on CXCL8 promoter activity is not provided. Authors should update the
methods section for all relevant experiments mentioned in the results section.

**Response:** We have incorporated the details of the reporter system used to
evaluate CXCL8 promoter activity into the revised manuscript (page 14, lines 483-
485).

I Suggest a grammar check for the entire manuscript. Some of the errors are listed

- Line 57 – correct word ‘Encubators’
- Line 105: Avoid ‘Remarkably’, ‘dramatically’
- Line 119: ‘Eliminate’ incorrect use
- Line 124: ‘final’ should be finally
- Line 219 – ‘Unbiased’, remove ‘an’.

Response: We have corrected these items in the revised manuscript (page 3, line 55; page 4, line 103; page 5, lines 122-123; page 5, line 130; page 8, line 239). Furthermore, the current resubmitted version has been refined by professional editors to increase the language quality.

Reviewer #2 (Remarks to the Author):

Response: Thank you very much for your valuable suggestion.

Reviewer #3 (Remarks to the Author):

Yang et al show that infection of A549 cells with EV-D68 induces expression and secretion of the proinflammatory cytokine CXCL8 or IL8. Silencing CXCL8 in multiple cell lines and in primary cells reduced EV-D68 replication, suggesting CXCL8 has a proviral effect. Replication was restored by exogenously added CXCL8. Targeting the CXCL8 signaling pathway, by downregulation of the receptor, or by known inhibitors, also inhibit viral replication. The authors attribute the proviral activity of CXCL8 to the activation of the MEK/ERK pathway, leading to hnRNPK translocation to the cytoplasm, that promotes translation from the 5’ UTR of the viral RNA. The authors also show that VP4 of EV-D68 is responsible for the CXCL8 upregulation through PI3K activation. Finally, they show that CXCL8 is also induced by other viruses, such as influenza and SARS-CoV-2, and it is also proviral. While the impact of CXCL8 in vitro in viral replication is clear from the data, the mechanistic data are not conclusive, and the biological impact in vivo remains unknown.

Response: We thank the reviewer for the careful reading of our manuscript and for the helpful comments and constructive criticism. Below are our point-by-point responses to the reviewer’s comments.

Specific comments

1. Fig 2j: Specificity of the inhibition of the 5’UTR by silencing CXCL8 should be shown by using an irrelevant 5’UTR. In addition, the impact of inhibitors in this assay was quite minimal, and perhaps CXCL8 signaling exerts its proviral activity at a different level from that of impacting 5’ viral UTR activity.

Response: As suggested by the reviewer, we investigated the effects of silencing CXCL8 on the activity of the HIV-1 5’LTR. The results indicated that the CXCL8 expression level does not influence the activity of the HIV-1 5’LTR, in contrast to

139 the relationship observed for the EV-D68 5'UTR (new Figure S6).
--We agree with the reviewer's comments. Although we observed a significant
promoting effect of the CXCL8 signalling pathway on the activity of the EV-D68 5'
UTR, the effect of silencing CXCL8 on 5' UTR activity was comparatively less
pronounced than its effect on viral replication. This discrepancy could be attributed
to the potential insensitivity of our luciferase reporter system in accurately
reflecting the actual viral infection dynamics. CXCL8 silencing attenuated 5' UTR
activity, which may have amplified its influence on viral replication. Furthermore,
the CXCL8 signalling pathway may participate in interference mechanisms in
addition to modulating viral 5'UTR activity during EV-D68 replication. We have
discussed this issue in detail in our revised manuscript (page 12, lines 398-402).

2. The experiments to define the pathways activated by CXCL8 that promote viral
replication are not easily interpretable. First, the inability of the virus to induce
MEK1/2-ERK1/2 signaling in a CXCL8 dependent manner might simply reflect that
viral replication (inhibited by silencing CXCL8) is required for MEK1/2-ERK1/2
signaling. Does addition of CXCL8 to the cells in the absence of viral replication
activate MEK1/2-ERK1/2 signaling? Same applies to translocation of hnRNPK.
Does addition of CXCL8 to the cells in the absence of viral replication results in
cytoplasmic translocation of hnRNPK, and is this dependent on MEK/ERK
signaling?

**Response:** Thank you very much for pointing this out. As suggested, we confirmed
that treatment with recombinant CXCL8 protein activated MAPK signalling (new
Figure S10a) and induced hnRNP K translocation in the absence of viral replication
(new Figure S11c). Moreover, inhibition of MAPK significantly suppressed the
CXCL8-mediated cytoplasmic translocation of hnRNP K (new Figure S11d).

The authors would need to somehow restore viral replication in the absence of
CXCL8 by activating the proposed downstream activities of CXCL8 in a CXCL8
independent manner (same way as they demonstrated that exogenous CXCL8
treatment compensates for silencing of CXCL8).

**Response:** We sincerely appreciate the reviewer's suggestion, and we conducted
the recommended experiments by using the MAPK agonist C16-PAF. The data
indicated significant restoration of EV-D68 replication in CXCL8-knockdown cells
upon treatment with C16-PAF (new Figure S10c-e).

3. While is quite interesting that other viruses activate CXCL8 expression and that
CXCL8 is also proviral, this raises concerns on whether the mechanism on how
CXCL8 enhances viral replication is actually the one proposed by the authors, as
the two other viruses do not share the same 5'UTR of EV-D68. Also, as in vivo
data are lacking on the impact of CXCL8 on viral replication and clearance, it is not
a given that CXCL8 is proviral in vivo. Perhaps CXCL8 is required for the
recruitment to the infected tissue of immune antiviral cells, and that its proviral
impact is shadowed by being required for a proper antiviral immune response in
in vivo by attracting neutrophils to the site of infection.

**Response:** --As suggested by the reviewers, we further investigated the

underlying mechanisms by which the CXCL8 signalling axis facilitates the
replication of other respiratory RNA viruses. First, we revealed that human
rhinovirus (HRV), a member of the enterovirus genus, induces CXCL8 expression
through an infection strategy similar to that of EV-D68 and relies on the CXCL8-
MAPK axis to induce cytoplasmic translocation of hnRNP K (new Figure 7e),
thereby increasing viral UTR activity (new Figure 7f) and promoting viral replication
(new Figure 7d). Intriguingly, influenza virus (IAV) induces CXCL8 expression via
its NS1 protein and depends on the CXCL8 axis to stimulate the translocation of
hnRNP K from the nucleus to the cytoplasm (new Figure 7j), facilitating the nuclear
export of viral M mRNA (new Figure 7k-n) and thereby augmenting IAV replication
(new Figure 7i). Our study revealed that different respiratory RNA viruses exploit
the CXCL8-MAPK signalling axis in a similar fashion to induce changes in the
subcellular localization of hnRNP K, thus promoting viral replication at various
stages. Targeting key effectors in this signalling pathway has potential applicability
for broad-spectrum inhibition of respiratory RNA viruses.

--The absence of the CXCL8 sequence in the mouse genome limits our ability to
conduct research using mouse infection models. To address this limitation, we
successfully demonstrated the beneficial impact of blocking the CXCL8 signalling
pathway on the inhibition of EV-D68 replication in human respiratory organoids
(new Figure 2j and 4k). Additionally, targeting cytokines, such as CXCL6/IL6, has
emerged as an effective strategy for combating severe viral infections. We concur
with the reviewers' meticulous comment stating that while EV-D68 induces CXCL8
expression to increase its replication capacity, it is imperative to also consider the
potential role of CXCL8 in tissue immune defence. As an alternative, we can
selectively target downstream effectors in the EV-D68-CXCL8 signalling pathway,
such as MAPK and hnRNP K, to retain the beneficial aspects of the immune
defence function of CXCL8. We have objectively discussed this issue as a
limitation of this study in the revised manuscript (page 14, lines 449-460).

**Reviewer #4 (Remarks to the Author):**

NCOMMS-24-14080-T review

The manuscript describes an interesting enhancement of EV-D68 replication when
signaling through the CXCL8 pathway is induced during virus infection. CXCL8
signaling increases the cytoplasmic amounts of hnRNP K which binds to EV-D68
viral RNA and enhances viral gene expression. The EV-D69 capsid protein VP4 is
also shown to mimic the function of ITAM and binds to syk kinase to increase
PI3K/AKT signaling which results in increased production of CXCL8. Eliminating
the interacting domain also eliminated VP4 activation of this pathway. Rhinovirus
VP4 encodes a similar interaction domain and its activation of CXCL8 signaling
was also dependent on the ITAM motif. Overall, the experiments are extensive and
well controlled. This is an important observation for picornavirus family viruses.

**Response:** We thank the reviewer for the careful reading and comments. We have
carefully considered these points and revised the manuscript accordingly. Specific
comments are addressed below.

Major points

Lines 193-196: This data is not shown or validated, which is really not appropriate
because the results very much dictate the rationale for going after hnRNPk as a
target.

**Response:** Thank you very much for this comment. After confirming that CXCL8
signalling had a positive effect on EV-D68 5'UTR activity, we investigated host
factors known to maintain EV-D68 5'UTR activity (including PCBP1, PCBP2, and
hnRNP K). Interestingly, while the expression level of CXCL8 did not affect the
ability of PCBP1 or PCBP2 to bind viral RNA (vRNA; new Figure S11a and S11b),
the knockdown of CXCL8 decreased the ability of EV-D68 to induce the
cytoplasmic accumulation of hnRNP K (Figure 5b) and hindered the interaction
between hnRNP K and vRNA (Figure 5c and 5d). As suggested by the reviewer,
we have incorporated this related information into the revised manuscript (page 7,
lines 210-214).

i'm a little less enthusiastic about the data in figures 7 d-h. this data shows some
dependence of other respiratory viruses on CXCL8 signaling, but the molecular
basis for that requirement is less clear. Is hnRNP also involved here or is an entirely
different mechanism mediating the CXCL8 effect here? this data seems a bit too
preliminary. In lines 354-356 the authors continue to discuss mechanisms by
which CXCL8 signaling might be aiding virus replication but influenza RNA
replication occurs in the nucleus so its difficult to see how the same mechanism
mediated by hnRNP could positively affect a cytoplasmic and a nuclear replicating
virus.

**Response:** We are very grateful for the constructive criticism provided by the
reviewers and have carefully considered the comments. Figure 7 has been
reorganized on the basis of the combined feedback from all the reviewers. In
response to the reviewer's concern regarding the CXCL8-mediated increase in IAV
replication, we conducted a series of experiments and confirmed that IAV infection
can induce the upregulation of CXCL8 (new Figure 7g), increase the activation of
the MAPK pathway (new Figure S14b), and result in the cytoplasmic translocation
of hnRNP K in a manner dependent on CXCL8 (new Figure 7j). From a mechanistic
perspective, recent studies by other teams have demonstrated that hnRNP K
directly binds to IAV RNA and promotes IAV replication (Thompson, M. G. et al.
*Nature Communications*, 2018, <https://doi.org/10.1038/s41467-018-04779-4>).
Cells infected with IAV mutants lacking hnRNP K binding ability exhibit nuclear
retention of IAV M mRNA. These published findings prompted our investigation
into the impact of IAV on the CXCL8-mediated nuclear export of viral mRNA. In
our system, we first observed that the knockdown of endogenous hnRNP K
resulted in the retention of IAV M mRNA within the nucleus (new Figure S16),
providing additional confirmation for the importance of hnRNP K in the M mRNA
export process. Subsequently, silencing CXCL8 disrupted the cytoplasmic
translocation of the hnRNP K protein induced by IAV infection and significantly
inhibited the nuclear export of IAV M mRNA (new Figure 7k-n). These results
suggest that, unlike EV-D68 and HRV, IAV relies on the CXCL8-MAPK axis to
induce the translocation of hnRNP K from the nucleus to the cytoplasm, thereby
promoting viral mRNA export. All the relevant data have been incorporated into the

revised manuscript (page 10, lines 302-315).

Minor points

Lines 57-63: this paragraph seems a bit odd. Adaption usually doesn't occur
quickly after infection and the first sentence seems more like literary writing as
opposed to scientific writing.

**Response:** As suggested by the reviewer, we have corrected the description in
the revised manuscript (page 3, lines 57-60).

Lines 86-87: What does "excessive background noise" mean?

**Response:** Our point is as follows: compared to the extensive variations in host
gene expression caused by multiple rounds of viral replication, only a limited
number of early response genes exhibit changes in expression as a result of viral
infection during the initial stages, which is a more advantageous condition for the
identification of crucial host factors involved in early stages of viral replication. To
address this issue, we have corrected this description in the revised manuscript
(page 4, lines 84-85).

Figure 2: the Figure 2C comparison is stated as being to uninfected cells but the
figure appears to compare infected cells with a control or CXCR8 specific siRNA.

**Response:** Thank you very much for pointing this out. We have corrected this in
the revised manuscript (page 4, line 94).

Figures 2i, 3c, 4d, 4f, 4k, 7c: is the y-axis in these figures in log10 scale? If so,
please label it as such.

**Response:** We have changed the axis labels in the revised Figures 2i, 3c, 4d, 4f,
4k, and 7c.

Line 188: viruses don't think so they can't intentionally do anything.

**Response:** We have changed the description in the revised manuscript (page 7,
line 204).

Line 201: this is not real time monitoring, as the cells were fixed and processed for
immunfluorescence analysis

**Response:** Thank you very much. We have corrected this issue in the revised
manuscript (page 7, line 219).

Figure 1 legend: what are the comparison groups that form the basis of the
statistical analysis? for example, is C being compared to mock infected or to 0h
infection?

**Response:** We appreciate the reviewer's comment. We have incorporated the
data for the mock-infected samples into new Figure 1c.

Figure 6i legend: indicate the MOI of infection the EV-D68 infections were
performed at. This seems like a rather strong effect to be mediated by virions
unless the infections were at an extremely high MOI.

**Response:** As suggested by the reviewer, we have incorporated this information
into the revised manuscript (page 32, line 1102).

REVIEWERS' COMMENTS

Reviewer #1 (Remarks to the Author):

The authors have addressed most of the comments to satisfaction through new
experiments and editing the manuscript. The authors have explained that in vivo animal
data is difficult for EV-68. However, it is not difficult for IAV. Nevertheless, they do provide
EV-68 data in the Organoid infection model, which is an acceptable alternative.

**Response:** We greatly appreciate your constructive suggestions and the recognition of
the potential importance of our work.

Reviewer #2 (Remarks to the Author):

I co-reviewed this manuscript with one of the reviewers who provided the listed reports.
This is part of the Nature Communications initiative to facilitate training in peer review and
to provide appropriate recognition for Early Career Researchers who co-review
manuscripts

**Response:** Thank you very much.

Reviewer #3 (Remarks to the Author):

The authors have responded adequately to the issues pointed by this reviewer

**Response:** Thank you very much.

Reviewer #4 (Remarks to the Author):

NCOMMS-24-14080A

The authors have responded appropriately to all the previous critiques and the manuscript
has been strengthened significantly.

**Response:** We greatly appreciate the recognition of the potential importance of our
discoveries by the reviewer and find the comments both insightful and helpful.